# Explicable Reward Design
# for Reinforcement Learning Agents

**Rati Devidze**[1]  **Goran Radanovic**[1]  **Parameswaran Kamalaruban**[2]  **Adish Singla**[1]

rdevidze@mpi-sws.org  gradanovic@mpi-sws.org  kparameswaran@turing.ac.uk  adishs@mpi-sws.org

[1]Max Planck Institute for Software Systems (MPI-SWS), Saarbrucken, Germany
[2]The Alan Turing Institute, London, UK

## Abstract

We study the design of *explicable* reward functions for a reinforcement learning agent while guaranteeing that an optimal policy induced by the function belongs to a set of target policies. By being explicable, we seek to capture two properties: (a) *informativeness* so that the rewards speed up the agent's convergence, and (b) *sparseness* as a proxy for ease of interpretability of the rewards. The key challenge is that higher informativeness typically requires dense rewards for many learning tasks, and existing techniques do not allow one to balance these two properties appropriately. In this paper, we investigate the problem from the perspective of discrete optimization and introduce a novel framework, EXPRD, to design explicable reward functions. EXPRD builds upon an informativeness criterion that captures the (sub-)optimality of target policies at different time horizons in terms of actions taken from any given starting state. We provide a mathematical analysis of EXPRD, and show its connections to existing reward design techniques, including potential-based reward shaping. Experimental results on two navigation tasks demonstrate the effectiveness of EXPRD in designing explicable reward functions.

## 1 Introduction

A reward function plays the central role during the learning/training process of a reinforcement learning (RL) agent. Given a "task" the agent is expected to perform (i.e., the desired learning outcome), there are typically many different reward specifications under which an optimal policy has the same performance guarantees on the task. This freedom in choosing the reward function, in turn, leads to the fundamental question of reward design: *What are different criteria that one should consider in designing a reward function for the agent, apart from the agent's final output policy?* [1–3].

One of the important criteria is *informativeness*, capturing that the rewards should speed up the agent's convergence [1–6]. For instance, a major challenge faced by an RL agent is because of delayed rewards during training; in the worst-case, the agent's convergence is slowed down exponentially w.r.t. the time horizon of delay [7]. In this case, we seek to design a new reward function that reduces this time horizon of delay while guaranteeing that any optimal policy induced by the designed function is also optimal under the original reward function [3]. The classical technique of potential-based reward shaping (when applied with appropriate state potentials) indeed allows us to reduce this time horizon of delay to 1; see [3, 8] and Section 2. With 1, it means that globally optimal actions for any state are also myopically optimal, thereby making the agent's learning process trivial.

While informativeness is an important criterion, it is not the only criterion to consider when designing rewards for many practical applications. Another natural criterion to consider is *sparseness* as a proxy for ease of interpretability of the rewards. There are several practical settings where sparseness and interpretability of rewards are important, as discussed next. The first motivating application is when

rewards are designed for human learners who are learning to perform sequential tasks, for instance, in pedagogical applications such as educational games [9], virtual reality-based training simulators [10, 11], and solving open-ended problems (e.g., block-based visual programming [12]). In this context, tasks can be challenging for novice learners and a teacher agent can assist these learners by designing explicable rewards associated with these tasks. The second motivating application is when rewards are designed for complex compositional tasks in the robotics domain that involve reward specifications in terms of logic, automata, or subgoals [13, 14]—these specifications induce a form of sparsity structure on the underlying reward function. The third motivating application is related to defense against reward-poisoning attacks in RL (see [15–19]) by designing structured and sparse reward functions that are easy to debug/verify. Beyond these practical settings, many naturally occurring reward functions in real-life tasks are inherently sparse and interpretable, further motivating the need to distill these properties in the automated reward design process. The key challenge is that higher informativeness typically requires dense rewards for many learning tasks – for instance, the above-mentioned potential-based shaped rewards that achieve a time horizon of 1 would require most of the states be associated with some real-valued reward (see Sections 2 and 4). To this end, an important research question that we seek to address is: *How to balance these two criteria of informativeness and sparseness in the reward design process while guaranteeing an optimality criterion on policies induced by the reward function?*

In this paper, we formalize the problem of designing *explicable* reward functions, focusing on the criteria of informativeness and sparseness. We investigate this problem from an expert/teacher's point of view who has full domain knowledge (in this case, an original reward function along with optimal policies induced by the original function), and seeks to design a new reward function for the agent—see Figure 1 and further discussion in Section 5 on expert-driven vs. agent-driven reward design. We tackle the problem from the perspective of discrete optimization and introduce a novel framework, EXPRD, to design reward functions. EXPRD allows us to appropriately balance informativeness and sparseness while guaranteeing that an optimal policy induced by the function belongs to a set of target policies. EXPRD builds upon an informativeness criterion that captures the (sub-)optimality of target policies at different time horizons from any given starting state. Our main contributions are:[1]

I. We formulate the problem of explicable reward functions to balance the two important criteria of informativeness and sparseness in the reward design process. (Sections 2 and 3.1)

II. We propose a novel optimization framework, EXPRD, to design reward functions. As part of this framework, we introduce a new criterion capturing informativeness of reward functions that is amenable to optimization techniques and is of independent interest. (Sections 3.2 and 3.3)

III. We provide a detailed mathematical analysis of EXPRD and show its connections to popular techniques, including potential-based reward shaping. (Sections 3.3 and 3.4)

IV. We provide a practical extension to apply our framework to large state spaces. We perform extensive experiments on two navigation tasks to demonstrate the effectiveness of EXPRD in designing explicable reward functions. (Sections 3.5 and 4)

## 2 Problem Setup

**Environment.** An environment is defined as a Markov Decision Process (MDP) $M := (\mathcal{S}, \mathcal{A}, T, \gamma, R)$, where the set of states and actions are denoted by $\mathcal{S}$ and $\mathcal{A}$ respectively. $T : \mathcal{S} \times \mathcal{S} \times \mathcal{A} \to [0, 1]$ captures the state transition dynamics, i.e., $T(s' \mid s, a)$ denotes the probability of landing in state $s'$ by taking action $a$ from state $s$. Here, $\gamma$ is the discounting factor. The underlying reward function is given by $R : \mathcal{S} \times \mathcal{A} \to [-R_{\max}, R_{\max}]$, for some $R_{\max} > 0$. We interchangeably represent the reward function by a vector $R \in \mathbb{R}^{|\mathcal{S}| \cdot |\mathcal{A}|}$, whose $(s|\mathcal{A}| + a)$-th entry is given by $R(s, a)$. We define the support of $R$ as $\mathrm{supp}(R) := \{s : s \in \mathcal{S}, R(s, a) \neq 0 \text{ for some } a \in \mathcal{A}\}$, and the $\ell_0$-norm of $R$ as $\|R\|_0 := |\mathrm{supp}(R)|$.

**Preliminaries and definitions.** We denote a stochastic policy $\pi : \mathcal{S} \to \Delta(\mathcal{A})$ as a mapping from a state to a probability distribution over actions, and a deterministic policy $\pi : \mathcal{S} \to \mathcal{A}$ as a mapping from a state to an action. For any policy $\pi$, the state value function $V_\infty^\pi$ and the action value function $Q_\infty^\pi$ in the MDP $M$ are defined as follows respectively: $V_\infty^\pi(s) = \mathbb{E}\left[\sum_{t=0}^\infty \gamma^t R(s_t, a_t) | s_0 = s, T, \pi\right]$ and $Q_\infty^\pi(s, a) = \mathbb{E}\left[\sum_{t=0}^\infty \gamma^t r_t | s_0 = s, a_0 = a, T, \pi\right]$. Further, the optimal value functions are given by $V_\infty^*(s) = \sup_\pi V_\infty^\pi(s)$ and $Q_\infty^*(s, a) = \sup_\pi Q_\infty^\pi(s, a)$. There always exists a

---

[1]Github repo: https://github.com/adishs/neurips2021_explicable-reward-design_code.

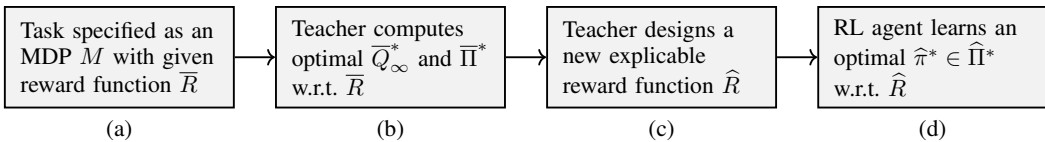

| Task specified as an MDP $M$ with given reward function $\overline{R}$ | $\rightarrow$ | Teacher computes optimal $\overline{Q}^*_\infty$ and $\overline{\Pi}^*$ w.r.t. $\overline{R}$ | $\rightarrow$ | Teacher designs a new explicable reward function $\widehat{R}$ | $\rightarrow$ | RL agent learns an optimal $\widehat{\pi}^* \in \widehat{\Pi}^*$ w.r.t. $\widehat{R}$ |
| (a) | | (b) | | (c) | | (d) |

Figure 1: Illustration of the explicable reward design problem in terms of a task specified through MDP $M$, an RL agent whose objective is to perform this task, and a teacher/expert whose objective is to help this RL agent. **(a)** MDP $M$ with a given reward function $\overline{R}$ specifying the task the RL agent is expected to perform; **(b)** The teacher computes optimal action value function $\overline{Q}^*_\infty$ along with the set of optimal policies $\overline{\Pi}^*$ w.r.t. $\overline{R}$; **(c)** The teacher designs a new explicable reward function $\widehat{R}$ for the RL agent; **(d)** The RL agent trains using the designed reward $\widehat{R}$ and outputs a policy $\widehat{\pi}^*$ from the set of optimal policies $\widehat{\Pi}^*$ w.r.t. $\widehat{R}$. Our framework designs an explicable reward function $\widehat{R}$ with three properties: *invariance*, *informativeness*, and *sparseness*; see main text for formal definitions of these properties.

deterministic stationary policy $\pi$ that achieves the optimal value function simultaneously for all $s \in \mathcal{S}$ [7, 20], and we denote all such deterministic optimal policies by the set $\Pi^* := \{\pi : \mathcal{S} \to \mathcal{A} \text{ s.t. } V^\pi_\infty(s) = V^*_\infty(s), \forall s \in \mathcal{S}\}$. From here onwards, we focus on deterministic policies unless stated otherwise. For any $\pi$ and $R$, we define the following quantities that capture the $\infty$-step (global) optimality gap and the 0-step (myopic) optimality gap of action $a$ at state $s$, respectively:

$$\delta^\pi_\infty(s,a) := Q^\pi_\infty(s,\pi(s)) - Q^\pi_\infty(s,a), \text{ and } \delta^\pi_0(s,a) := Q^\pi_0(s,\pi(s)) - Q^\pi_0(s,a), \forall s \in \mathcal{S}, a \in \mathcal{A},$$

where $Q^\pi_0(s,a) = R(s,a)$ is the 0-step action value function of policy $\pi$. The $\delta^\pi_\infty(s,a)$ values are same for all $\pi \in \Pi^*$, and we denote it by $\delta^*_\infty(s,a) = V^*_\infty(s) - Q^*_\infty(s,a)$; however, this is not the case with $\delta^\pi_0(s,a)$ values in general. For any state $s \in \mathcal{S}$ and a set of policies $\Pi$, we define $\Pi_s := \{a : a = \pi(s), \pi \in \Pi\}$. Then, we have that $\delta^*_\infty(s,a) = 0, \forall s \in \mathcal{S}, a \in \Pi^*_s$.

**Explicable reward design.** Figure 1 presents an illustration of the explicable reward design problem that we formalize below. A task is specified as an MDP $M$ with a given goal-based reward function $\overline{R}$ where $\overline{R}$ has non-zero rewards only on goal states $\mathcal{G} \subseteq \mathcal{S}$, i.e., $\overline{R}(s,a) = 0, \forall s \in \mathcal{S}\backslash\mathcal{G}, a \in \mathcal{A}$. Many naturally occurring tasks (see Section 1 for motivating applications) are goal-based and challenging for learning an optimal policy when the state space $\mathcal{S}$ is very large. In this paper, we study the following explicable reward design problem from an expert/teacher's point of view: Given $\overline{R}$ and the corresponding optimal policy set $\overline{\Pi}^*$ w.r.t. $\overline{R}$ as the input, the teacher designs a new reward function $\widehat{R}$ with criteria of *informativeness* and *sparseness* while guaranteeing an *invariance* requirement (these properties are formalized in Section 3). Informally, the invariance requirement is that any optimal policy learned using the new reward $\widehat{R}$ belongs to the optimal policy set $\overline{\Pi}^*$ induced by $\overline{R}$.[2]

**Typical techniques for reward design and issues.** Given a set of important states (subgoals) in the environment, one could design a handcrafted reward function $\widehat{R}_{\text{CRAFT}}$ by assigning non-zero reward values only to these states. Even though this simple approach produces a reward function with a specified sparsity level, it often fails to satisfy the invariance requirement. In particular, there are some well-known "reward bugs" that can arise in this approach and mislead the agent into learning sub-optimal policies (see [2, 3]). In the seminal work [3], the authors introduced the potential-based reward shaping (PBRS) method to alleviate this issue. The reward function produced by the PBRS method with optimal value function $\overline{V}^*_\infty$ under $\overline{R}$ as the potential function is defined as follows:

$$\widehat{R}_{\text{PBRS}}(s,a) := \overline{R}(s,a) + \gamma \sum_{s' \in \mathcal{S}} T(s' \mid s,a) \cdot \overline{V}^*_\infty(s') - \overline{V}^*_\infty(s). \tag{1}$$

The set of optimal policies $\widehat{\Pi}^*$ induced by $\widehat{R}_{\text{PBRS}}$ is exactly equal to the set of optimal policies $\overline{\Pi}^*$ induced by $\overline{R}$ since $\widehat{\delta}^\pi_\infty(s,a) = \overline{\delta}^*_\infty(s,a)$ for all $\pi \in \overline{\Pi}^*$ [3]. In addition, for any state $s \in \mathcal{S}$, globally optimal actions $\overline{\Pi}^*_s \subseteq \mathcal{A}$ under $\overline{R}$ are also myopically optimal under $\widehat{R}_{\text{PBRS}}$ since $\widehat{\delta}^\pi_0(s,a) = \overline{\delta}^*_\infty(s,a)$ for all $\pi \in \overline{\Pi}^*$ [3, 8] – this leads to a dramatic speed-up in the learning process. However, the potential-based reward shaping produces dense reward function which is less interpretable (see Section 4).

---

[2]In the rest of the paper, the quantities defined corresponding to $R := \overline{R}$ are denoted by an *overline*, e.g., the optimal policy set by $\overline{\Pi}^*$ and the $\infty$-step optimality gaps by $\overline{\delta}^*_\infty$; the quantities defined corresponding to $R := \widehat{R}$ are denoted by a *widehat*, e.g., the optimal policy set by $\widehat{\Pi}^*$.

# 3 Our Reward Design Framework ExPRD

In Sections 3.1, 3.2, and 3.3, we propose an optimization formulation and a greedy solution for the explicable reward design problem. In Section 3.4, we provide a theoretical analysis of our greedy solution. In Section 3.5, we provide a practical extension to apply our framework to large state spaces.

## 3.1 Discrete Optimization Formulation

Given $\overline{R}$ and the corresponding optimal policy set $\overline{\Pi}^*$, we systematically develop a discrete optimization framework (ExPRD) to design an explicable reward function $\widehat{R}$ (see Figure 1).

**Sparseness, informativeness, and invariance.** The sparseness of the reward function $\widehat{R}$ is captured by $\mathrm{supp}(\widehat{R})$. In Section 3.2, we formalize an informativeness criterion $I(\widehat{R})$ of $\widehat{R}$ that captures how hard/easy it is to learn an optimal behavior induced by $\widehat{R}$. We explicitly enforce the invariance requirement (see Section 2) for the new reward $\widehat{R}$ by choosing a set of candidate policies $\Pi^\dagger \subseteq \overline{\Pi}^*$, and satisfying the following (Bellman-optimality) conditions:

$$Q_\infty^{\pi^\dagger}(s,a) \;=\; \widehat{R}(s,a) + \gamma \sum_{s' \in \mathcal{S}} T(s'|s,a) \cdot Q_\infty^{\pi^\dagger}(s',\pi^\dagger(s')), \quad \forall a \in \mathcal{A}, s \in \mathcal{S}, \pi^\dagger \in \Pi^\dagger \quad \text{(C.1)}$$

$$Q_\infty^{\pi^\dagger}(s,\pi^\dagger(s)) \;\geq\; Q_\infty^{\pi^\dagger}(s,a) + \overline{\delta}_\infty^*(s), \quad \forall a \in \mathcal{A}\backslash\overline{\Pi}_s^*, s \in \mathcal{S}, \pi^\dagger \in \Pi^\dagger, \quad \text{(C.2)}$$

where $\overline{\delta}_\infty^*(s) := \min_{a \in \mathcal{A}\backslash\overline{\Pi}_s^*} \overline{\delta}_\infty^*(s,a), \forall s \in \mathcal{S}$.[3] The above conditions guarantee that any optimal policy induced by $\widehat{R}$ is also optimal under $\overline{R}$, i.e., $\Pi^\dagger \subseteq \widehat{\Pi}^* \subseteq \overline{\Pi}^*$. Here, the set $\Pi^\dagger \subseteq \overline{\Pi}^*$ is used to reduce the number of constraints. Note that for the potential-based shaped reward $\widehat{R}_{\mathrm{PBRS}}$, we have $\widehat{\Pi}^* = \overline{\Pi}^*$.

**Maximizing informativeness for a given set of important states.** When a domain expert provides us a set of important states (subgoals) in the environment [21–24], we want to use this set in a principled way to design a reward $\widehat{R}$, while avoiding the "reward bugs" that can arise from hand-crafted rewards $\widehat{R}_{\mathrm{CRAFT}}$. To this end, for any given set of subgoals $\mathcal{Z} \subseteq \mathcal{S}\backslash\mathcal{G}$, we optimize the informativeness criterion $I(R)$ while satisfying the invariance requirement:

$$g(\mathcal{Z}) \;:=\; \max_{R:\mathrm{supp}(R)\subseteq\mathcal{Z}\cup\mathcal{G}} \quad I(R)$$

$$\text{subject to} \quad \text{conditions (C.1)} - \text{(C.2) with } \widehat{R} \text{ replaced by } R \text{ hold} \quad \text{(P1)}$$
$$|R(s,a)| \;\leq\; R_{\max}, \; \forall s \in \mathcal{S}, a \in \mathcal{A}.$$

Let $R^{(\mathcal{Z})}$ denote the $R$ that maximizes $g(\mathcal{Z})$. Let $\mathcal{R} \subseteq \mathbb{R}^{|\mathcal{S}|\cdot|\mathcal{A}|}$ be a constraint set on $R$ that captures only the conditions (C.1) − (C.2) and the $R_{\max}$ bound.

**Jointly finding subgoals along with maximizing informativeness.** Based on (P1), we propose the following discrete optimization formulation that allows us to select a set of important states (of size $B$) and design a reward function that maximizes informativeness automatically:

$$\max_{\mathcal{Z}:\mathcal{Z}\subseteq\mathcal{S}\backslash\mathcal{G},|\mathcal{Z}|\leq B} g(\mathcal{Z}). \quad \text{(P2)}$$

We can incorporate prior knowledge about the quality of subgoals using a set function $D : 2^{\mathcal{S}} \to \mathbb{R}$ (we assume $D$ to be a submodular function [25]). Finally, the full ExPRD formulation is given by:

$$\max_{\mathcal{Z}:\mathcal{Z}\subseteq\mathcal{S}\backslash\mathcal{G},|\mathcal{Z}|\leq B} g(\mathcal{Z}) + \lambda \cdot D(\mathcal{Z}\cup\mathcal{G}), \quad \text{for some } \lambda \geq 0. \quad \text{(P3)}$$

We study the problems (P1), (P2), and (P3) in the following subsections.

---

[3]Note that the true action values $\overline{Q}_\infty^*$ are used in the conditions (C.1) − (C.2) to obtain the terms $\overline{\delta}_\infty^*(s,a)$, $\mathcal{A}\backslash\overline{\Pi}_s^*$, and $\Pi^\dagger$. However, when we only have an approximate estimate of $\overline{Q}_\infty^*$, we can adapt (C.1) − (C.2) appropriately with approximate versions of $\overline{\delta}_\infty^*(s,a)$, $\mathcal{A}\backslash\overline{\Pi}_s^*$, and $\Pi^\dagger$.

## 3.2 Informativeness Criterion

Understanding the informativeness of a reward function is an important problem, and several works have investigated it [4, 5, 26–28]. Our goal is to define an informativeness criterion that is amenable to optimization techniques. As noted in Section 2, for any policy $\pi \in \overline{\Pi}^*$, 0-step and $\infty$-step optimality gaps induced by $\widehat{R}_{\text{PBRS}}$ are all equal to $\infty$-step optimality gaps induced by $\overline{R}$, i.e., $\widehat{\delta}_0^\pi(s,a) = \widehat{\delta}_\infty^\pi(s,a) = \overline{\delta}_\infty^*(s,a)$. For any reward function $R$, one could ask how much these two quantities could differ, and even consider the intermediate cases between 0-step and $\infty$-step optimality. Inspired by the $h$-step optimality notions studied in [4, 26], we define the $h$-step action value function of any policy $\pi$ as $Q_h^\pi(s,a) = \mathbb{E}\left[\sum_{t=0}^h \gamma^t R(s_t, a_t)|s_0 = s, a_0 = a, T, \pi\right]$, and it satisfies the following recursive relationship: $Q_h^\pi(s,a) = R(s,a) + \gamma \sum_{s' \in \mathcal{S}} T(s'|s,a) \cdot Q_{h-1}^\pi(s', \pi(s'))$.

Let $\mathcal{H}$ be a set of horizons for which we want to maximize informativeness. For any policy $\pi$ and reward function $R$, we define the following quantity that captures the $h$-step optimality gap of action $a$ at state $s$: $\delta_h^\pi(s,a) := Q_h^\pi(s, \pi(s)) - Q_h^\pi(s,a), \forall s \in \mathcal{S}, a \in \mathcal{A}, h \in \mathcal{H}$. Later, in the proof of Proposition 2, we show that $\delta_h^\pi(s,a)$ is linear in $R$, i.e., $\delta_h^\pi(s,a) = \langle w_{h;(s,a)}, R \rangle$ for some vector $w_{h;(s,a)} \in \mathbb{R}^{|\mathcal{S}| \cdot |\mathcal{A}|}$. Interestingly, the following proposition states that, for any policy $\pi \in \overline{\Pi}^*$ and any $h$, the $h$-step optimality gap induced by $\widehat{R}_{\text{PBRS}}$ given in (1) is equal to the $\infty$-step optimality gap induced by $\overline{R}$:

**Proposition 1.** *The goal-based reward function $\overline{R}$, and the potential-based shaped reward function $\widehat{R}_{\text{PBRS}}$ given in (1) satisfy the following: $\widehat{\delta}_h^\pi(s,a) = \overline{\delta}_\infty^*(s,a), \forall s \in \mathcal{S}, a \in \mathcal{A}, \pi \in \overline{\Pi}^*, h \in \mathcal{H}$.*

Let $\ell : \mathbb{R} \to \mathbb{R}$ be a monotonically non-decreasing concave function. Then, based on the $h$-step optimality gaps, we define the informativeness criterion of the reward $R$ as follows:

$$I_\ell(R) := \sum_{\pi^\dagger \in \Pi^\dagger} \sum_{h \in \mathcal{H}} \sum_{s \in \mathcal{S}} \sum_{a \in \mathcal{A} \setminus \overline{\Pi}_s^*} \ell(\delta_h^{\pi^\dagger}(s,a)).$$

From here onwards, we let $I$ be $I_\ell$ in the problem (P1). As an example for $\ell$, we consider the negated hinge loss given by $\ell_{\text{hg}}(\delta(s,a)) := -\max(0, \overline{\delta}_\infty^*(s,a) - \delta(s,a))$. By Proposition 1, we have that $I_{\ell_{\text{hg}}}(\widehat{R}_{\text{PBRS}}) = 0$, and $I_{\ell_{\text{hg}}}(R) \leq 0$ for any other $R$, i.e., $\widehat{R}_{\text{PBRS}}$ achieves the maximum value of $I_{\ell_{\text{hg}}}$.

## 3.3 Iterative Greedy Algorithm

First, we show that the problem (P1) can be efficiently solved using the standard concave optimization methods to find $R^{(\mathcal{Z})}$ for any given $\mathcal{Z} \subseteq \mathcal{S} \setminus \mathcal{G}$:

**Proposition 2.** *For any given $\mathcal{Z} \subseteq \mathcal{S} \setminus \mathcal{G}$, the problem (P1) is a concave optimization problem in $R \in \mathbb{R}^{|\mathcal{S}| \cdot |\mathcal{A}|}$ with linear constraints. Further, the feasible set of the problem (P1) is non-empty.*

Then, inspired by the Forward Stepwise Selection method from [29], we propose an iterative greedy solution (see Algorithm 1) to solve the problems (P2) and (P3). To compute the incremental gain at each step, we would need to solve the concave optimization problem (P1) for different values of $\mathcal{Z}$. The problem (P1) has $|\mathcal{S}| \cdot |\mathcal{A}|$ optimization variables and $\mathcal{O}(|\mathcal{S}| \cdot |\mathcal{A}| \cdot |\Pi^\dagger| \cdot |\mathcal{H}|)$ constraints.

---

**Algorithm 1** Iterative Greedy Algorithm for EXPRD

---

1: **Input:** MDP $\overline{M} := (\mathcal{S}, \mathcal{A}, T, \gamma, \overline{R})$, $\overline{\delta}_\infty^*(s,a)$ values, sets $\overline{\Pi}^*, \Pi^\dagger, \mathcal{G}, \mathcal{H}$, sparsity budget $B$
2: **Initialize:** $\mathcal{Z}_0 \leftarrow \emptyset$
3: **for** $k = 1, 2, \ldots, B$ **do**
4: $\quad z_k \leftarrow \arg\max_{z \in \mathcal{S} \setminus \mathcal{Z}_{k-1}} g(\mathcal{Z}_{k-1} \cup \{z\}) + \lambda \cdot D(\mathcal{Z}_{k-1} \cup \mathcal{G} \cup \{z\}) - g(\mathcal{Z}_{k-1}) - \lambda \cdot D(\mathcal{Z}_{k-1} \cup \mathcal{G})$
5: $\quad \mathcal{Z}_k \leftarrow \mathcal{Z}_{k-1} \cup \{z_k\}$
6: **Output:** $\mathcal{Z}_B$ and the corresponding optimal reward function $R^{(\mathcal{Z}_B)}$.

---

## 3.4 Theoretical Analysis

Here, we provide guarantees for the solution returned by our Algorithm 1. Below, we give an overview of the main technical ideas, and leave a detailed discussion along with proofs in the

Appendix. For some $\mu \geq 0$, let $I_\ell^{\text{reg}}(R) := I_\ell(R) - \mu \|R\|_2^2$ be the regularized informativeness criterion. We define a normalized set function $f : 2^{\mathcal{S}} \to \mathbb{R}$ as follows:

$$f(\mathcal{Z}) = \max_{R:\text{supp}(R) \subseteq \mathcal{Z} \cup \mathcal{G}, R \in \mathcal{R}} (I_\ell^{\text{reg}}(R) - I_\ell^{\text{reg}}(R^{(\emptyset)})) + \lambda \cdot (D(\mathcal{Z} \cup \mathcal{G}) - D(\mathcal{G})), \quad (2)$$

where $R^{(\emptyset)} = \arg\max_{R:\text{supp}(R) \subseteq \mathcal{G}, R \in \mathcal{R}} I_\ell^{\text{reg}}(R)$. Note that the regularized variant ($I_\ell$ replaced by $I_\ell^{\text{reg}}$) of the optimization problem (P3) is equivalent to $\max_{\mathcal{Z}:\mathcal{Z} \subseteq \mathcal{S} \backslash \mathcal{G}, |\mathcal{Z}| \leq B} f(\mathcal{Z})$. For a given sparsity budget $B$, let $\mathcal{Z}_B^{\text{Greedy}}$ be the set selected by our Algorithm 1 and $\mathcal{Z}_B^{\text{OPT}}$ be the optimal set that maximizes the regularized variant of problem (P3). The corresponding $f$ values of these sets are denoted by $f_B^{\text{Greedy}}$ and $f_B^{\text{OPT}}$ respectively; in the following, we are interested in comparing these two values. The problem (P3) is closely related to the subset selection problem studied in [29] with a twist of an additional constraint set $\mathcal{R}$ (see the discussion after (P1)), making the theoretical analysis more challenging. Inspired by the analysis in [29], we need to prove a weak form of submodularity [25, 30] for $f$ (since $D$ is already a submodular function, we need to prove this for the case when $\lambda = 0$). To this end, we require the regularized informativeness criterion $I_\ell^{\text{reg}}$ to satisfy certain structural assumptions. First, we define the restricted strongly concavity and restricted smoothness notions of a function that are used in our analysis.

**Definition 1** (Restricted Strong Concavity, Restricted Smoothness [31]). *A function $\mathcal{L} : \mathbb{R}^{|\mathcal{S}| \cdot |\mathcal{A}|} \to \mathbb{R}$ is said to be restricted strong concave with parameter $m_\Omega$ and restricted smooth with parameter $M_\Omega$ on a domain $\Omega \subset \mathbb{R}^{|\mathcal{S}| \cdot |\mathcal{A}|} \times \mathbb{R}^{|\mathcal{S}| \cdot |\mathcal{A}|}$ if for all $(x, y) \in \Omega$:*

$$-\frac{m_\Omega}{2} \|y - x\|_2^2 \geq \mathcal{L}(y) - \mathcal{L}(x) - \langle \nabla \mathcal{L}(x), y - x \rangle \geq -\frac{M_\Omega}{2} \|y - x\|_2^2.$$

*For any integer $k$, we define the following two sets: $\Omega_k := \{(x, y) : \|x\|_0 \leq k, \|y\|_0 \leq k, \|x - y\|_0 \leq k, x, y \in \mathcal{R}\}$, and $\tilde{\Omega}_k := \{(x, y) : \|x\|_0 \leq k, \|y\|_0 \leq k, \|x - y\|_0 \leq 1, x, y \in \mathcal{R}\}$. Let $m_k := m_{\Omega_k}$ and $M_k := M_{\Omega_k}$ (similarly we define $\tilde{m}_k$ and $\tilde{M}_k$).*

When there is no $R \in \mathcal{R}$ constraint in (2), the following assumption on the regularized informativeness criterion is sufficient to prove the weak submodularity of $f$ [29]:

**Assumption 1.** *The regularized informativeness criterion $I_\ell^{\text{reg}}$ is $m_{2B+|\mathcal{G}|}$-restricted strongly concave and $M_{2B+|\mathcal{G}|}$-restricted smooth on $\Omega_{2B+|\mathcal{G}|}$.*

However, due to the additional $R \in \mathcal{R}$ constraint, we need to enforce further requirements on $I_\ell^{\text{reg}}$ formally captured in Assumption 2 provided in the Appendix; here, we discuss these requirements informally. Let $\mathcal{Z}$ be any set such that $\mathcal{Z} \subseteq \mathcal{S} \backslash \mathcal{G}$, and $\nabla I_\ell^{\text{reg}}(R^{(\mathcal{Z})})$ be the gradient of the regularized informativeness criterion at the optimal reward $R^{(\mathcal{Z})}$. Then, we need to ensure the following: (i) the $\ell_2$-norm of the projection of $\nabla I_\ell^{\text{reg}}(R^{(\mathcal{Z})})$ on $(\mathcal{Z} \cup \mathcal{G})$ is upper-bounded, captured by $d_{\max}^{\text{opt}}$; (ii) the $\ell_2$-norm of the projection of $\nabla I_\ell^{\text{reg}}(R^{(\mathcal{Z})})$ on any $j \in \mathcal{S} \backslash (\mathcal{Z} \cup \mathcal{G})$ is lower-bounded, captured by $d_{\min}^{\text{non}}$; and (iii) the components of the optimal reward $R^{(\mathcal{Z})}$ outside $(\mathcal{Z} \cup \mathcal{G})$ do not lie in the boundary of $\mathcal{R}$, captured by $\kappa$. Then, by using Assumption 1 and Assumption 2 (see Appendix), we prove the weak submodularity of $f$. Finally, by applying Theorem 3 from [29], we obtain the following theorem:

**Theorem 1.** *Let $I_\ell^{\text{reg}}$ satisfies Assumption 1 and Assumption 2 requirements. Then, we have $f_B^{\text{Greedy}} \geq (1 - e^{-\gamma}) f_B^{\text{OPT}}$, where $\gamma = \frac{\kappa \cdot m_{2B+|\mathcal{G}|}}{M_{2B+|\mathcal{G}|}} \cdot \frac{(d_{\min}^{\text{non}})^2}{(d_{\max}^{\text{opt}})^2 + (d_{\min}^{\text{non}})^2}$.*

We provide Assumption 2 and a detailed proof of the theorem in the Appendix.

### 3.5 Extension to Large State Spaces using State Abstractions

This section presents an extension of our EXPRD framework that is scalable to large state spaces by leveraging the techniques from state abstraction literature [32–34]. We use an abstraction $\phi : \mathcal{S} \to \mathcal{X}_\phi$, which is a mapping from high-dimensional state space $\mathcal{S}$ to a low-dimensional latent space $\mathcal{X}_\phi$. Let $\phi^{-1}(x) := \{s \in \mathcal{S} : \phi(s) = x\}, \forall x \in \mathcal{X}_\phi$, and $\overline{M} := (\mathcal{S}, \mathcal{A}, T, \gamma, \overline{R})$. We propose the following pipeline:

1. By using $\overline{M}$ and $\phi$, we construct an abstract MDP $\overline{M}_\phi = (\mathcal{X}_\phi, \mathcal{A}, T_\phi, \gamma, \overline{R}_\phi)$ as follows, $\forall x, x' \in \mathcal{X}_\phi, a \in \mathcal{A}$: $T_\phi(x'|x, a) = \frac{1}{|\phi^{-1}(x)|} \sum_{s \in \phi^{-1}(x)} \sum_{s' \in \phi^{-1}(x')} T(s'|s, a)$, and $\overline{R}_\phi(x, a) = \frac{1}{|\phi^{-1}(x)|} \sum_{s \in \phi^{-1}(x)} \overline{R}(s, a)$. We compute the set of optimal policies $\overline{\Pi}_\phi^*$ for the MDP $\overline{M}_\phi$.

2. We run our EXPRD framework on $\overline{M}_\phi$ with $\Pi^\dagger = \overline{\Pi}_\phi^*$, and the resulting reward is denoted $\widehat{R}_\phi$.

3. We define the reward function $\widehat{R}$ on the state space $\mathcal{S}$ as follows: $\widehat{R}(s, a) = \widehat{R}_\phi(\phi(s), a)$.

By assuming certain structural conditions on $\phi$ formalized in the Appendix, we can show that any optimal policy induced by the above reward $\widehat{R}$ acts nearly optimal w.r.t. $\overline{R}$. This pipeline can be extended to continuous state space as well, similar to [34–36]. We provide more details in the Appendix.

## 4 Experimental Evaluation

In this section, we evaluate EXPRD on two environments: ROOMSNAVENV (Section 4.1) and LINEKEYNAVENV (Section 4.2). ROOMSNAVENV corresponds to a navigation task in a grid-world where the agent has to learn a policy to quickly reach the goal location in one of four rooms, starting from an initial location. Even though this environment has a small state space, it provides a very rich and an intuitive problem setting to validate different reward design techniques, and variants of ROOMSNAVENV have been used extensively in the literature [14, 21, 22, 37–40]. LINEKEYNAVENV corresponds to a navigation task in a one-dimensional space where the agent has to first pick the key and then reach the goal. The agent's location in this environment is represented as a point on a line segment. Given the large state space representation, it is computationally challenging to apply the reward design technique from Section 3.3 and we use the state abstraction-based extension of our framework from Section 3.5. This environment is inspired by variants of navigation tasks in the literature where an agent needs to perform sub-tasks [3, 41]. We give an overview of main results here, and provide a more detailed description of the setup and additional results in the Appendix.

### 4.1 Evaluation on ROOMSNAVENV

**ROOMSNAVENV (Figure 2).** We represent the environment as an MDP with $\mathcal{S}$ states each corresponding to cells in the grid-world indicating the agent's current location (shown as "blue-circle"). Goal (shown as "green-star") is located at the top-right corner cell. The agent can take four actions given by $\mathcal{A} := \{$"up", "left", "down", "right"$\}$. An action takes the agent to the neighbouring cell represented by the direction of the action; however, if there is a wall (shown as "brown-segment"), the agent stays at the current location. Furthermore, when an agent takes an action $a \in \mathcal{A}$, there is $p_{\text{rand}}$ probability that an action $a' \in \mathcal{A} \setminus \{a\}$ will be executed instead of $a$. In addition to these walls, there are a few terminal walls (shown as "thick-red-segment") that terminates the

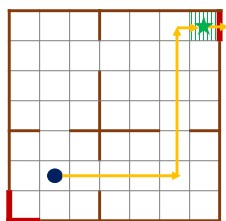

Figure 2: ROOMSNAVENV

episode—at the bottom-left corner cell, "left" and "down" actions terminate; at the top-right corner cell, "right" action terminates. The agent gets a reward of $R_{\text{max}}$ after it has navigated to the goal and then takes a "right" action (i.e., only one state-action pair has a reward); note that this action also terminates the episode. The reward is 0 for all other state-action pairs and there is a discount factor $\gamma$. This MDP has $|\mathcal{S}| = 49$ and $|\mathcal{A}| = 4$; we set $p_{\text{rand}} = 0.1$, $R_{\text{max}} = 10$, and $\gamma = 0.95$ in our evaluation.

**Techniques evaluated.** We consider the following baselines: (i) $\widehat{R}_{\text{ORIG}} := \overline{R}$, which simply represents default reward function, (ii) $\widehat{R}_{\text{PBRS}}$ obtained via the PBRS technique with the optimal value function $\overline{V}_\infty^*$ w.r.t. $\overline{R}$ (see Section 2), (iii) $\widehat{R}_{\text{CRAFT}}$ that we design manually (see Section 2 and description below), and (iv) $\widehat{R}_{\text{PBRS-CRAFT}(B=5)}$ obtained via the PBRS technique with the optimal value function w.r.t. $\widehat{R}_{\text{CRAFT}}$ instead of $\overline{V}_\infty^*$ [42].[4] To design $\widehat{R}_{\text{CRAFT}}$, we first hand-crafted a set function $D$ that assigns scores to the states in the MDP, e.g., the scores are higher for the four entry points in the rooms. In general, one could learn such $D$ automatically using the techniques from [21–24]—see full details about $D$ in the Appendix. Then, for a fixed budget $B$, we pick the top $B$ states according to the scoring by $D$ and assign a reward of $+1$ for optimal actions and $-1$ for others. For the evaluation, we use $B = 5$ and denote the function as $\widehat{R}_{\text{CRAFT}(B=5)}$. Note that apart from $B$ states, $\widehat{R}_{\text{CRAFT}(B=5)}$ also has a reward assigned for the goal state taken from $\overline{R}$.

---

[4]The reward shaping method in [42] is based on the PBRS technique and leads to dense reward functions. However, their method is more practical as it does not require solving the original task w.r.t. $\overline{R}$.

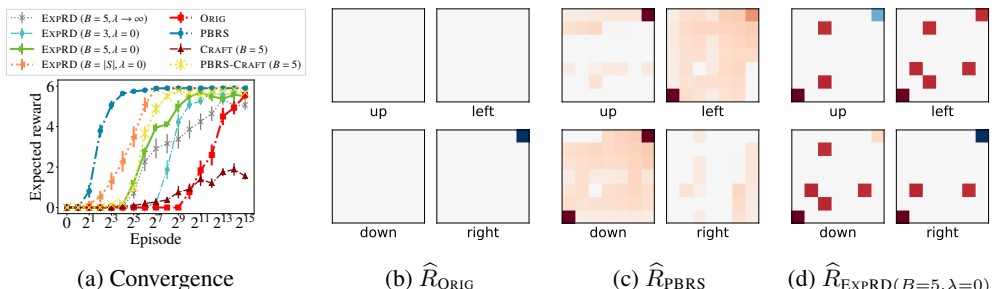

| (a) Convergence | (b) $\widehat{R}_{\text{ORIG}}$ | (c) $\widehat{R}_{\text{PBRS}}$ | (d) $\widehat{R}_{\text{EXPRD}(B=5,\lambda=0)}$ |
|---|---|---|---|

Figure 3: Results for ROOMSNAVENV. **(a)** shows convergence in performance of the agent w.r.t. training episodes. Here, performance is measured as the expected reward per episode computed using $\overline{R}$; note that the x-axis is exponential in scale. **(b-d)** visualize the designed reward functions $\widehat{R}_{\text{ORIG}}$, $\widehat{R}_{\text{PBRS}}$, and $\widehat{R}_{\text{EXPRD}(B=5,\lambda=0)}$. These plots illustrate reward values for all combinations of $\mathcal{S} \times \mathcal{A}$ shown as four $7 \times 7$ grids corresponding to different actions. Blue color represents positive reward, red color represents negative reward, and the magnitude of the reward is indicated by color intensity. As an example, consider "right" action grid for $\widehat{R}_{\text{ORIG}}$ in **(b)** where the dark blue color in the corner indicates the goal. To increase the color contrast, we clipped rewards in the range $[-4, +4]$ for this visualization even though the designed rewards are in the range $[-10, +10]$. See Section 4.1 for details.

The reward functions $\widehat{R}_{\text{EXPRD}}$ designed by our EXPRD framework are parameterized by budget $B$ and hyperparameter $\lambda$. For $\lambda$, we consider two extreme settings: (a) $\lambda = 0$ where the problem (P3) reduces to (P2), and (b) $\lambda \to \infty$ where the problem (P3) reduces to (P1) corresponding to the reward design with subgoals pre-selected by the function $D$. We use the same function $D$ that we used for $\widehat{R}_{\text{CRAFT}}$ above. For budget $B$, we consider values from $\{3, 5, |S|\}$. In particular, we evaluate the following reward functions: $\widehat{R}_{\text{EXPRD}(B=5,\lambda\to\infty)}$, $\widehat{R}_{\text{EXPRD}(B=3,\lambda=0)}$, $\widehat{R}_{\text{EXPRD}(B=5,\lambda=0)}$, and $\widehat{R}_{\text{EXPRD}(B=|S|,\lambda=0)}$. For the evaluation in this section, we use the following parameter choices for EXPRD: $\mathcal{H} = \{1, 4, 8, 16, 32\}$, $\ell$ is the negated hinge loss $\ell_{\text{hg}}$, and $\Pi^{\dagger}$ contains only one policy from $\overline{\Pi}^{*}$.

**Results.** We use standard Q-learning method for the agent with a learning rate $0.5$ and exploration factor $0.1$ [7]. During training, the agent receives rewards based on $\widehat{R}$, however, is evaluated based on $\overline{R}$. A training episode ends when the maximum steps (set to $50$) is reached or an agent's action terminates the episode. All the results are reported as average over $40$ runs and convergence plots show mean with standard error bars. The convergence behavior in Figure 3a demonstrates the effectiveness of the reward functions designed by our EXPRD framework.[5] Note that $\widehat{R}_{\text{CRAFT}(B=5)}$ leads to sub-optimal behavior due to "reward bugs" (see Section 2), whereas $\widehat{R}_{\text{EXPRD}(B=5,\lambda\to\infty)}$ fixes this issue using the same set of subgoals. EXPRD leads to good performance even without domain knowledge (i.e., when $\lambda = 0$), e.g., the performance corresponding to $\widehat{R}_{\text{EXPRD}(B=3,\lambda=0)}$ is comparable to that of $\widehat{R}_{\text{EXPRD}(B=5,\lambda\to\infty)}$. The visualizations of $\widehat{R}_{\text{ORIG}}$, $\widehat{R}_{\text{PBRS}}$, and $\widehat{R}_{\text{EXPRD}(B=5,\lambda=0)}$ in Figures 3b, 3c, and 3d highlight the trade-offs in terms of sparseness and interpretability of the reward functions. The reward function $\widehat{R}_{\text{EXPRD}(B=5,\lambda=0)}$ designed by our EXPRD framework provides a good balance in terms of convergence performance while maintaining high sparseness. Additional visualizations and results are provided in the Appendix.

### 4.2 Evaluation on LINEKEYNAVENV

**LINEKEYNAVENV (Figure 4).** We represent the environment as an MDP with $\mathcal{S}$ states corresponding to the agent's status comprising of the current location (shown as "blue-circle" and is a point x in $[0, 1]$) and a binary flag whether the agent has acquired a key (shown as "cyan-bolt"). Goal (shown as "green-star") is available in locations on the segment $[0.9, 1]$, and the key is available in locations on the segment $[0.1, 0.2]$. The agent can take three actions given by $\mathcal{A} := \{\text{"left"}, \text{"right"}, , \text{"pick"}\}$. "pick" action does not change the agent's location, however, when

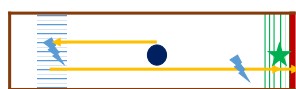

Figure 4: LINEKEYNAVENV

---

[5] As we discussed in Sections 1 and 2, $\widehat{R}_{\text{PBRS}}$ designed using $\overline{V}^{*}_{\infty}$ makes the agent's learning process trivial.

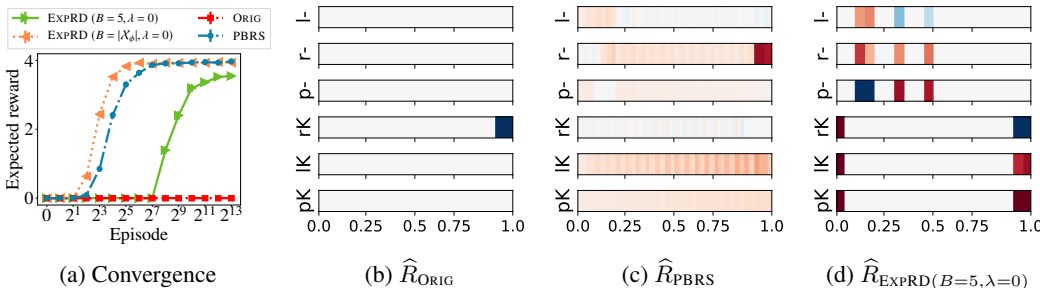

| (a) Convergence | (b) $\widehat{R}_{\text{ORIG}}$ | (c) $\widehat{R}_{\text{PBRS}}$ | (d) $\widehat{R}_{\text{EXPRD}(B=5,\lambda=0)}$ |

Figure 5: Results for LINEKEYNAVENV. **(a)** shows convergence in performance of the agent w.r.t. training episodes. Here, performance is measured as the expected reward per episode computed using $\overline{R}$. **(b-d)** visualize the designed reward functions $\widehat{R}_{\text{ORIG}}$, $\widehat{R}_{\text{PBRS}}$, and $\widehat{R}_{\text{EXPRD}(B=5,\lambda=0)}$. These plots illustrate reward values for all combination of triplets, i.e., agent's location on the segment $[0.0, 1.0]$ (shown as horizontal bar), agent's status whether it has acquired key or not (indicated as 'K' or '-'), and three actions (indicated as 'l' for "left", 'r' for "right", 'p' for "pick"). We use a color representation similar to Figure 3, and we clipped rewards in the range $[-3, +3]$ to increase the color contrast for this visualization. As an example, consider 'rK' bar for $\widehat{R}_{\text{ORIG}}$ in **(b)** where the dark blue color on the segment $[0.9, 1]$ indicate the locations with goal. See Section 4.2 for details.

executed in locations with availability of the key, the agent acquires the key; if agent already had a key, the action does not affect the status. A move action of "left" or "right" takes the agent from the current location in the direction of move with the dynamics of the final location captured by two hyperparameters $(\Delta_{a,1}, \Delta_{a,2})$; for instance, with current location x and action "left", the new location x′ is sampled uniformly among locations from $(x - \Delta_{a,1} - \Delta_{a,2})$ to $(x - \Delta_{a,1} + \Delta_{a,2})$. Similar to ROOMSNAVENV, the agent's move action is not applied if the new location crosses the wall, and there is $p_{\text{rand}}$ probability of a random action. The agent gets a reward of $R_{\text{max}}$ after it has navigated to the goal locations after acquiring the key and then takes a "right" action; note that this action also terminates the episode. The reward is 0 elsewhere and there is a discount factor $\gamma$. We set $p_{\text{rand}} = 0.1$, $R_{\text{max}} = 10$, $\gamma = 0.95$, $\Delta_{a,1} = 0.075$ and $\Delta_{a,2} = 0.01$.

**Techniques evaluated.** The baseline $\widehat{R}_{\text{ORIG}} := \overline{R}$ represents the default reward function. We evaluate the variants of $\widehat{R}_{\text{PBRS}}$ and $\widehat{R}_{\text{EXPRD}}$ using an abstraction. For a given hyperparameter $\alpha \in (0, 1)$, the set of possible locations $X$ are obtained by $\alpha$-level discretization of the line segment from 0.0 to 1.0, leading to a $1/\alpha$ set of locations. For the abstraction $\phi$ associated with this discretization [43], the abstract MDP $\overline{M}_\phi$ (see Section 3.5) has $|\mathcal{X}_\phi| = 2/\alpha$ and $|\mathcal{A}| = 3$. We use $\alpha = 0.05$. We compute the optimal state value function in the abstract MDP $\overline{M}_\phi$, lift it to the original state space via $\phi$, and use the lifted value function as the potential to design $\widehat{R}_{\text{PBRS}}$ [35]. We follow the pipeline in Section 3.5 to design $\widehat{R}_{\text{EXPRD}}$ − in the subroutine, we run EXPRD on $\overline{M}_\phi$ for a budget $B = 5$ and a full budget $B = |\mathcal{X}_\phi|$; we set $\lambda = 0$. For other parameters ($\mathcal{H}$, $\ell$, and $\Pi^\dagger$), we use the same choices as in Section 4.1.

**Results.** The agent uses Q-learning method in the original MDP $\overline{M}$ by using a fine-grained discretization of the state space; rest of the method's parameters are same as in Section 4.1. All the results are reported as average over 40 runs and convergence plots show mean with standard error bars. Figure 5a demonstrates that all three designed reward functions—$\widehat{R}_{\text{PBRS}}$, $\widehat{R}_{\text{EXPRD}(B=5,\lambda=0)}$, $\widehat{R}_{\text{EXPRD}(B=|\mathcal{X}_\phi|,\lambda=0)}$—substantially improves the convergence, whereas the agent is not able to learn under $\widehat{R}_{\text{ORIG}}$. Based on the visualizations in Figures 5b, 5c, and 5d, $\widehat{R}_{\text{EXPRD}(B=5,\lambda=0)}$ provides a good balance between convergence and sparseness. Interestingly, $\widehat{R}_{\text{EXPRD}(B=5,\lambda=0)}$ assigned a high positive reward for the "pick" action when the agent is in the locations with key (see 'p-' bar in Figure 5d).

## 5 Related Work

**Potential-based reward shaping.** Introduced in the seminal work of [3], potential-based reward shaping is one of the most well-studied reward design technique (see [8, 14, 37, 38, 40, 44, 45, 45–48]). As we discussed in Section 2, the shaped reward function $\widehat{R}_{\text{PBRS}}$ is obtained by modifying $\overline{R}$ using a state-dependent potential function. The technique preserves a strong invariance property: a

policy $\pi$ is optimal under $\widehat{R}_{\text{PBRS}}$ *iff* it is optimal under $\overline{R}$. Furthermore, when using the optimal value-function $\overline{V}_\infty^*$ under $\overline{R}$ as the potential function, the shaped rewards achieve the maximum possible informativeness as per the notion we use in EXPRD. To balance informativeness and sparseness, our framework EXPRD can be seen as a relaxation of the potential-based shaping in the following ways: (i) EXPRD provides a guarantee on preserving a weaker invariance property whereby an optimal policy under $\widehat{R}_{\text{EXPRD}}$ is also optimal under $\overline{R}$; (ii) EXPRD finds $\widehat{R}_{\text{EXPRD}}$ that maximizes informativeness under hard constraints of preserving this weaker policy-invariant property and a given spareness-level.

**Optimization-based techniques for reward design.** Beyond potential-based shaping, we can formulate reward design as an optimization problem [15–19]. In particular, optimization-based techniques for reward design are popularly used in data poisoning attacks where an attacker's goal is to minimally perturb the original reward function to force the agent into executing a target policy chosen by the attacker [17–19]. Our EXPRD framework builds on the optimization framework of [17–19]. The key novelty of EXPRD is that we optimize for informativeness of the reward function under a sparseness constraint, which makes our problem formulation much more challenging.

**Agent-driven reward design.** An important categorization of reward design techniques is based on who is designing the rewards and what domain knowledge is available. Agent-driven reward design techniques involve a reinforcement learning method where an agent self-designs its own rewards during the training process, with the objective of improving the exploration and speeding up the convergence [6, 49–53]. These agent-driven techniques use a wide-variety of ideas such as designing intrinsic rewards based on exploration bonus [49, 50, 54], designing rewards using some additional domain knowledge [51], and using credit assignment to create intermediate rewards [6, 52].

**Expert-driven reward design.** In contrast to agent-driven techniques, we have expert-driven reward design techniques where an expert/teacher with full domain knowledge can design a reward function for the agent [1, 3, 14–19, 48]. Our EXPRD framework falls into the category of teacher-driven reward design. The above-mentioned techniques of potential-based reward shaping and optimization-based techniques can be seen as expert-driven reward design techniques; however, the distinction between expert-driven and agent-driven techniques can be blurry at times when one uses an expert-driven technique with minimal domain knowledge (e.g., when using approximate potentials [3]).

**Reward automatas, landmark-based rewards, and subgoal discovery.** Our EXPRD framework is also connected to techniques that specify rewards using higher-level abstract representations of the environment including symbolic automata and landmarks [13, 14, 37, 40, 55, 56]. In recent works [13, 14, 55, 56], potential-based reward shaping technique has been used with automata-based rewards to design interpretable and informative rewards. While similar in the overall objective, our work is technically quite different and our proposed optimization framework to reward design can be seen as complementary to these works. Another relevant line of work focuses on automatic discovery of subgoals in the environment [21–24] – these works are complementary and useful as subroutines in our framework by providing a prior knowledge about which states are important for assigning rewards.

## 6    Conclusions

We developed a novel optimization framework, EXPRD, to design explicable reward functions in which we can appropriately balance informativeness and sparseness in the reward design process. As part of the framework, we introduced a new criterion capturing informativeness of reward functions that is of independent interest. The mathematical analysis of EXPRD shows connections of our framework to the popular reward-design techniques, and provides theoretical underpinnings of expert-driven explicable reward design. Importantly, EXPRD allows one to go beyond using a potential function for principled reward design, and provides a general recipe for developing an optimization-based reward design framework with different structural constraints. We also provided a practical extension to apply our framework in environments with large state spaces via state abstractions.

To make our framework more scalable, we plan to investigate alternate formulations of the reward design problem that avoids enumerating all the constraints explicitly (see Section 3). There are several promising directions for future work, including but not limited to the following: (a) using a combination of our optimization-based reward design technique with automata-driven rewards as well as other structured rewards, (b) extending our framework for agent-driven reward design, (c) applying our framework in a transfer setting using techniques from [42, 57], and (d) investigating the usage of our informativeness criterion for discovering subgoals.

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
