# A    List of Appendices

In this section, we give a brief description of the content provided in the appendices of the paper.

- Appendix B provides proofs for Propositions 1 and 2. (Sections 3.2 and 3.3)
- Appendix C provides additional details and proofs for the theoretical analysis. (Section 3.4)
- Appendix D provides additional details and proofs for using state abstractions. (Section 3.5)
- Appendix E provides additional results for ROOMSNAVENV. (Section 4.1)
- Appendix F provides additional results for LINEKEYNAVENV. (Section 4.2)

# B    Proofs for Propositions 1 and 2 (Sections 3.2 and 3.3)

## B.1    Proof of Proposition 1

**Proposition 1.** *The goal-based reward function $\overline{R}$, and the potential-based shaped reward function $\widehat{R}_{\mathrm{PBRS}}$ given in (1) satisfy the following: $\widehat{\delta}_h^\pi(s, a) = \overline{\delta}_\infty^*(s, a), \forall s \in \mathcal{S}, a \in \mathcal{A}, \pi \in \overline{\Pi}^*, h \in \mathcal{H}.$*

*Proof.* Consider any optimal policy $\pi \in \overline{\Pi}^*$, $s \in \mathcal{S}$, $a \in \mathcal{A}$, and $h \in \mathcal{H}$. The $\infty$-step optimality gap induced by $\overline{R}$ is $\overline{\delta}_\infty^*(s, a) = \overline{V}_\infty^*(s) - \overline{Q}_\infty^*(s, a)$, and the $h$-step optimality gap induced by $\widehat{R}_{\mathrm{PBRS}}$ is $\widehat{\delta}_h^\pi(s, a) = \widehat{Q}_h^\pi(s, \pi(s)) - \widehat{Q}_h^\pi(s, a)$. In the following, we express the two terms of $\widehat{\delta}_h^\pi$ in terms of $\overline{V}_\infty^*$ and $\overline{Q}_\infty^*$.

**The term $\widehat{Q}_h^\pi(s, \pi(s))$ for any $\pi \in \overline{\Pi}^*$.**    We show that $\widehat{Q}_h^\pi(s, \pi(s)) = 0$ for any non-negative integer $h$ by using mathematical induction. First ($h = 0$ case), we consider the 0-step optimal action value function:

$$
\begin{aligned}
\widehat{Q}_0^\pi(s, \pi(s)) &= \widehat{R}_{\mathrm{PBRS}}(s, \pi(s)) \\
&= \overline{R}(s, \pi(s)) + \gamma \sum_{s' \in \mathcal{S}} T(s' \mid s, \pi(s)) \overline{V}_\infty^*(s') - \overline{V}_\infty^*(s) \\
&= \overline{Q}_\infty^*(s, \pi(s)) - \overline{V}_\infty^*(s) \\
&= \overline{V}_\infty^*(s) - \overline{V}_\infty^*(s) \\
&= 0.
\end{aligned}
$$

Now assume that $\widehat{Q}_{h-1}^\pi(s, \pi(s)) = 0$. Then, consider the $h$-step optimal action value function:

$$
\begin{aligned}
\widehat{Q}_h^\pi(s, \pi(s)) &= \widehat{R}_{\mathrm{PBRS}}(s, \pi(s)) + \gamma \sum_{s' \in \mathcal{S}} T(s' \mid s, \pi(s)) \widehat{Q}_{h-1}^\pi(s', \pi(s)) \\
&= \widehat{R}_{\mathrm{PBRS}}(s, \pi(s)) + 0 \\
&= 0.
\end{aligned}
$$

Thus, by mathematical induction, we have that $\widehat{Q}_h^\pi(s, \pi(s)) = 0$ for any non-negative integer $h$.

**The term $\widehat{Q}_h^\pi(s, a)$ for any $a \in \mathcal{A}$.**    Consider the $h$-step optimal action value function:

$$
\begin{aligned}
\widehat{Q}_h^\pi(s, a) &= \widehat{R}_{\mathrm{PBRS}}(s, a) + \gamma \sum_{s' \in \mathcal{S}} T(s' \mid s, a) \widehat{Q}_{h-1}^\pi(s', \pi(s')) \\
&= \widehat{R}_{\mathrm{PBRS}}(s, a) + 0 \\
&= \overline{R}(s, a) + \gamma \sum_{s' \in \mathcal{S}} T(s' \mid s, a) \overline{V}_\infty^*(s') - \overline{V}_\infty^*(s) \\
&= \overline{Q}_\infty^*(s, a) - \overline{V}_\infty^*(s).
\end{aligned}
$$

Finally, by combining these two terms, we get:

$$
\widehat{\delta}_h^\pi(s, a) = \widehat{Q}_h^\pi(s, \pi(s)) - \widehat{Q}_h^\pi(s, a) = \overline{V}_\infty^*(s) - \overline{Q}_\infty^*(s, a) = \overline{\delta}_\infty^*(s, a).
$$

$\square$

## B.2 Proof of Proposition 2

**Proposition 2.** *For any given $\mathcal{Z} \subseteq \mathcal{S} \backslash \mathcal{G}$, the problem (P1) is a concave optimization problem in $R \in \mathbb{R}^{|\mathcal{S}| \cdot |\mathcal{A}|}$ with linear constraints. Further, the feasible set of the problem (P1) is non-empty.*

*Proof.* We write the problem (P1) explicitly as follows:

$$\max_{R} \sum_{\pi^\dagger \in \Pi^\dagger} \sum_{h \in \mathcal{H}} \sum_{s \in \mathcal{S}} \sum_{a \in \mathcal{A} \backslash \overline{\Pi}_s^*} \ell(\delta_h^{\pi^\dagger}(s,a)) \tag{3}$$

subject to $R(s,a) = 0, \forall s \in \mathcal{S} \backslash \{\mathcal{Z} \cup \mathcal{G}\}, a \in \mathcal{A}$ (4)

$$Q_\infty^{\pi^\dagger}(s,a) = R(s,a) + \gamma \sum_{s' \in \mathcal{S}} T(s'|s,a) Q_\infty^{\pi^\dagger}(s', \pi^\dagger(s')), \forall s \in \mathcal{S}, a \in \mathcal{A}, \pi^\dagger \in \overline{\Pi}^\dagger \tag{5}$$

$$Q_\infty^{\pi^\dagger}(s, \pi^\dagger(s)) \geq Q_\infty^{\pi^\dagger}(s,a) + \overline{\delta}_\infty^*(s), \forall s \in \mathcal{S}, a \in \mathcal{A} \backslash \overline{\Pi}_s^*, \pi^\dagger \in \overline{\Pi}^\dagger \tag{6}$$

$$Q_0^{\pi^\dagger}(s,a) = R(s,a), \forall s \in \mathcal{S}, a \in \mathcal{A}, \pi^\dagger \in \overline{\Pi}^\dagger \tag{7}$$

$$Q_h^{\pi^\dagger}(s,a) = R(s,a) + \gamma \sum_{s' \in \mathcal{S}} T(s'|s,a) Q_{h-1}^{\pi^\dagger}(s', \pi(s')), \forall s \in \mathcal{S}, a \in \mathcal{A}, h \in \mathcal{H}, \pi^\dagger \in \overline{\Pi}^\dagger$$
$$\tag{8}$$

$$\delta_h^{\pi^\dagger}(s,a) = Q_h^\pi(s, \pi^\dagger(s)) - Q_h^{\pi^\dagger}(s,a), \forall s \in \mathcal{S}, a \in \mathcal{A}, h \in \mathcal{H}, \pi^\dagger \in \overline{\Pi}^\dagger \tag{9}$$

$$|R(s,a)| \leq R_{\max}, \forall s \in \mathcal{S}, a \in \mathcal{A} \tag{10}$$

In the following, we show that the above problem is a concave optimization problem (the objective is concave and the constraints are linear) by writing it in the matrix form as follows:

$$\max_{R \in \mathbb{R}^{|\mathcal{S}| \cdot |\mathcal{A}|}} \sum_{\pi^\dagger \in \Pi^\dagger} \sum_{h \in \mathcal{H}} \sum_{s \in \mathcal{S}} \sum_{a \in \mathcal{A} \backslash \overline{\Pi}_s^*} \ell\left(\left\langle w_{h;(s,a)}^{\pi^\dagger}, R \right\rangle\right)$$

$$\text{subject to } A \cdot R \succeq b,$$

for some vectors $w_{h;(s,a)}^{\pi^\dagger}, b \in \mathbb{R}^{|\mathcal{S}| \cdot |\mathcal{A}|}$, and some matrix $A \in \mathbb{R}^{|\mathcal{S}| \cdot |\mathcal{A}| \times |\mathcal{S}| \cdot |\mathcal{A}|}$.

**Notation.** We mainly follow the notation from [58]. Given a deterministic policy $\pi : \mathcal{S} \to \mathcal{A}$, we define the transition matrix $T_\pi \in \mathbb{R}^{|\mathcal{S}| \cdot |\mathcal{A}| \times |\mathcal{S}| \cdot |\mathcal{A}|}$ induced by $\pi$ as follows:

$$[T_\pi]_{(s,a),(s',a')} := \begin{cases} T(s'|s,a), & \text{if } a' = \pi(s') \\ 0, & \text{otherwise.} \end{cases}$$

Also, for any $s \in \mathcal{S}$, we define $\text{Id}_\pi(s) \in \mathbb{R}^{|\mathcal{A}| \times |\mathcal{A}|}$ as follows:

$$[\text{Id}_\pi(s)]_{:,a} := \begin{cases} 1, & \text{if } a = \pi(s) \\ 0, & \text{otherwise.} \end{cases}$$

Then, we define $\text{Id}_\pi \in \mathbb{R}^{|\mathcal{S}| \cdot |\mathcal{A}| \times |\mathcal{S}| \cdot |\mathcal{A}|}$ as a block diagonal matrix with block size of $|\mathcal{A}| \times |\mathcal{A}|$, and $\text{Id}_\pi(s)$ as the $s^{\text{th}}$ diagonal block, $\forall s \in \mathcal{S}$. We define the diagonal matrix $L_{\overline{\Pi}^*} \in \mathbb{R}^{|\mathcal{S}| \cdot |\mathcal{A}| \times |\mathcal{S}| \cdot |\mathcal{A}|}$, whose $(s,a)^{\text{th}}$ diagonal entry is given by:

$$[L_{\overline{\Pi}^*}]_{(s,a),(s,a)} := \begin{cases} 0, & \text{if } a \in \overline{\Pi}_s^* \\ 1, & \text{otherwise.} \end{cases}$$

We define the diagonal matrix $L_{\mathcal{Z}} \in \mathbb{R}^{|\mathcal{S}| \cdot |\mathcal{A}| \times |\mathcal{S}| \cdot |\mathcal{A}|}$, whose $(s,a)^{\text{th}}$ diagonal entry is given by:

$$[L_{\mathcal{Z}}]_{(s,a),(s,a)} := \begin{cases} 0, & \text{if } s \in \mathcal{Z} \\ 1, & \text{otherwise.} \end{cases}$$

Let $e_i \in \mathbb{R}^{|\mathcal{S}| \cdot |\mathcal{A}|}$ be a vector having 1 only in the $i^{\text{th}}$ entry, and 0 elsewhere. Let $\overline{\delta}_\infty^* \in \mathbb{R}^{|\mathcal{S}| \cdot |\mathcal{A}|}$ be a vector such that its $(s,a)^{\text{th}}$ entry is given by $\left[\overline{\delta}_\infty^*\right]_{(s,a)} = \overline{\delta}_\infty^*(s), \forall a \in \mathcal{A}$. Let $\mathbf{1} \in \mathbb{R}^{|\mathcal{S}| \cdot |\mathcal{A}|}$ be a vector of all ones. Let $\text{Id} \in \mathbb{R}^{|\mathcal{S}| \cdot |\mathcal{A}| \times |\mathcal{S}| \cdot |\mathcal{A}|}$ be the identity matrix.

**Bound constraint.** The bound constraint in Eq. (10) can be written as follows:

$$R_{\max} \cdot \mathbf{1} \ \succeq \ R \ \succeq \ -R_{\max} \cdot \mathbf{1}.$$

The above is linear inequality in $R$.

**Sparsity constraint.** The sparsity constraint in Eq. (4) can be written as follows:

$$L_{\mathcal{Z}} R \ = \ 0.$$

The above is linear equality in $R$.

**Global optimality constraints.** The recursive form of the action value function $Q_\infty^\pi(s, a) = R(s, a) + \gamma \sum_{s' \in \mathcal{S}} T(s' \mid s, a) Q_\infty^\pi(s', \pi(s'))$ can be written in the matrix form as follows:

$$Q_\infty^\pi \ = \ R + \gamma T_\pi Q_\infty^\pi \ \implies \ Q_\infty^\pi \ = \ (\mathrm{Id} - \gamma T_\pi)^{-1} R.$$

Then, the global optimality constraints in Eq. (6) can be written as follows, for all $\pi^\dagger \in \overline{\Pi}^\dagger$:

$$(\mathrm{Id}_{\pi^\dagger} - \mathrm{Id}) \, Q_\infty^{\pi^\dagger} \ \succeq \ L_{\overline{\Pi}^*} \overline{\delta}_\infty^* \ \implies \ (\mathrm{Id}_{\pi^\dagger} - \mathrm{Id}) \, (\mathrm{Id} - \gamma T_{\pi^\dagger})^{-1} R \ \succeq \ L_{\overline{\Pi}^*} \overline{\delta}_\infty^*.$$

The above is linear inequality in $R$.

**Information $I_\ell(R)$ is concave in $R$.** For $h = 0$, $Q_0^\pi(s, a) = R(s, a)$ can be written as follows:

$$Q_0^\pi \ = \ R.$$

For $h = 1$, $Q_1^\pi(s, a) = R(s, a) + \gamma \sum_{s' \in \mathcal{S}} T(s' \mid s, a) Q_0^\pi(s', \pi(s'))$ can be written as follows:

$$Q_1^\pi \ = \ R + \gamma T_\pi Q_0^\pi \ = \ (\mathrm{Id} + \gamma T_\pi) R.$$

For $h = 2$, $Q_2^\pi(s, a) = R(s, a) + \gamma \sum_{s' \in \mathcal{S}} T(s' \mid s, a) Q_1^\pi(s', \pi(s'))$ can be written as follows:

$$Q_2^\pi \ = \ R + \gamma T_\pi Q_1^\pi \ = \ \left( \mathrm{Id} + \gamma T_\pi + \gamma^2 T_\pi T_\pi \right) R.$$

For any $h$, $Q_h^\pi(s, a) = R(s, a) + \gamma \sum_{s' \in \mathcal{S}} T(s' \mid s, a) Q_{h-1}^\pi(s', \pi(s'))$ can be written as follows:

$$Q_h^\pi \ = \ \left( \mathrm{Id} + \gamma T_\pi + \gamma^2 T_\pi^{(2)} + \cdots + \gamma^h T_\pi^{(h)} \right) R,$$

where $T_\pi^{(h)} = \underbrace{T_\pi T_\pi \cdots T_\pi}_{h-\text{times}}$. Then, we can write $\delta_h^\pi(s, a) = Q_h^\pi(s, \pi(s)) - Q_h^\pi(s, a)$ as follows:

$$\delta_h^\pi(s, a) \ = \ \left\langle (\mathrm{Id}_\pi - \mathrm{Id}) \left( \mathrm{Id} + \gamma T_\pi + \gamma^2 T_\pi^{(2)} + \cdots + \gamma^h T_\pi^{(h)} \right) R, e_{(s,a)} \right\rangle,$$

i.e., $\delta_h^\pi(s, a)$ is linear in $R$ for every $s \in \mathcal{S}$, and $a \in \mathcal{A}$. From the above equation, one can easily show that $\delta_h^\pi(s, a) = \left\langle w_{h;(s,a)}^{\pi^\dagger}, R \right\rangle$, where $w_{h;(s,a)}^{\pi^\dagger} := \rho_{h;(s,\pi^\dagger(s))}^{\pi^\dagger} - \rho_{h;(s,a)}^{\pi^\dagger}$. Since $\ell : \mathbb{R} \to \mathbb{R}$ is monotonically non-decreasing concave function, we have that $\ell \circ \delta_h^\pi(s, a)$ is concave [59]. From the fact that the sum of concave functions is concave, $I_\ell(R)$ is concave in $R$.

In summary, for the problem (P1), the objective is concave and the constraints are of linear form $(A \cdot R \succeq b)$. Thus, (P1) is a concave optimization problem.

**Feasibility.** One can easily verify that the original reward function $\overline{R}$ satisfies all the constraints in (4)-(10) of the sparse reward shaping formulation for any $\mathcal{Z}$, i.e., $\overline{R}$ is a feasible solution. Furthermore, when $\mathcal{Z} = \mathcal{S} \backslash \mathcal{G}$, the potential-based shaped reward function $\widehat{R}_{\mathrm{PBRS}}$ given in (1) satisfies all the constraints in (4)-(10) of the sparse reward shaping formulation. $\qquad \square$

# C Additional Details and Proofs for Theoretical Analysis (Section 3.4)

First, we define the submodularity and weak submodularity notions of a normalized set function, which are used in the proof of Theorem 1.

**Definition 2** (Submodularity [60]). *Let $g : 2^{\mathcal{V}} \to \mathbb{R}$ be a normalized set function ($g(\emptyset) = 0$). $g$ is submodular if for all $\mathcal{W} \subseteq \mathcal{V}$ and $j, k \in \mathcal{V} \backslash \mathcal{W}$:*

$$g(\mathcal{W} \cup \{k\}) - g(\mathcal{W}) \geq g(\mathcal{W} \cup \{j, k\}) - g(\mathcal{W} \cup \{j\}).$$

**Definition 3** (Weak Submodularity [30]). *Let $\mathcal{Y}, \mathcal{X} \subset \mathcal{V}$ be two disjoint sets, and $g : 2^{\mathcal{V}} \to \mathbb{R}$ be a normalized set function. The submodularity ratio of $\mathcal{X}$ with respect to $\mathcal{Y}$ is given by*

$$\gamma_{\mathcal{X}, \mathcal{Y}} := \frac{\sum_{j \in \mathcal{Y}} (g(\mathcal{X} \cup \{j\}) - g(\mathcal{X}))}{g(\mathcal{X} \cup \mathcal{Y}) - g(\mathcal{X})}. \tag{11}$$

*The submodularity ratio of a set $\mathcal{W}$ with respect to an integer $k$ is given by*

$$\gamma_{\mathcal{W}, k} := \min_{\mathcal{X}, \mathcal{Y} : \mathcal{X} \cap \mathcal{Y} = \emptyset, \mathcal{X} \subseteq \mathcal{W}, |\mathcal{Y}| \leq k} \gamma_{\mathcal{X}, \mathcal{Y}}.$$

*Let $\gamma > 0$. We call a function $\gamma$-weakly submodular at a set $\mathcal{W}$ and an integer $k$ if $\gamma_{\mathcal{W}, k} \geq \gamma$.*

A set function $g : 2^{\mathcal{V}} \to \mathbb{R}$ is called monotone if and only if $g(\mathcal{X}) \leq g(\mathcal{Y})$ for all $\mathcal{X} \subseteq \mathcal{Y}$.

For any $x \in \mathbb{R}^{|\mathcal{S}| \cdot |\mathcal{A}|}$ and $\mathcal{U} \subseteq \mathcal{S}$, $x_{\mathcal{U}}$ is defined as $x_{\mathcal{U}}(j, a) = x(j, a), \forall a \in \mathcal{A}$ when $j \in \mathcal{U}$, and $x_{\mathcal{U}}(j, a) = 0, \forall a \in \mathcal{A}$ otherwise. For any $j \in \mathcal{S}$, $e_j \in \mathbb{R}^{|\mathcal{S}| \cdot |\mathcal{A}|}$ is defined as $e_j(j', a) = 1, \forall a \in \mathcal{A}$ when $j' = j$, and $e_j(j', a) = 0, \forall a \in \mathcal{A}$ otherwise. The following assumption captures the additional requirements on the regularized informativeness criterion $I_\ell^{\text{reg}}$:

**Assumption 2.** *Let $\mathcal{Z}$ be any set such that $\mathcal{Z} \subseteq \mathcal{S} \backslash \mathcal{G}$. The regularized informativeness criterion $I_\ell^{\text{reg}}$ satisfies the following:*

- $\left\| \nabla I_\ell^{\text{reg}}(R^{(\mathcal{Z})})_{(\mathcal{Z} \cup \mathcal{G})} \right\|_2 \leq d_{\max}^{\text{opt}}$,

- $\left\| \nabla I_\ell^{\text{reg}}(R^{(\mathcal{Z})})_j \right\|_2 \geq d_{\min}^{\text{non}}, \forall j \in \mathcal{S} \backslash (\mathcal{Z} \cup \mathcal{G})$,

- $\left\| \nabla I_\ell^{\text{reg}}(R^{(\mathcal{Z})})_j \right\|_\infty \leq d_{\max}^{\text{non}}, \forall j \in \mathcal{S} \backslash (\mathcal{Z} \cup \mathcal{G})$, *and*

- $\exists \kappa \leq 1$ *such that* $\forall j \in \mathcal{S} \backslash (\mathcal{Z} \cup \mathcal{G}) : R^{(\mathcal{Z})} \pm \kappa \cdot \frac{d_{\max}^{\text{non}}}{\overline{M}_{|\mathcal{Z}| + |\mathcal{G}| + 1}} \cdot e_j \in \mathcal{R}$.

## C.1 Proof of Theorem 1

Let $\mathcal{Z} \subseteq \mathcal{S} \backslash \mathcal{G}$. Consider the set function $f : 2^{\mathcal{S}} \to \mathbb{R}_+$ defined in (2):

$$f(\mathcal{Z}) = \max_{R : \text{supp}(R) \subseteq \mathcal{Z} \cup \mathcal{G}, R \in \mathcal{R}} (I_\ell^{\text{reg}}(R) - I_\ell^{\text{reg}}(R^{(\emptyset)})) + \lambda \cdot (D(\mathcal{Z} \cup \mathcal{G}) - D(\mathcal{G})),$$

where $R^{(\emptyset)} = \arg \max_{R : \text{supp}(R) \subseteq \mathcal{G}, R \in \mathcal{R}} I_\ell^{\text{reg}}(R)$. Note that $f$ is a normalized, monotone set function. For a given sparsity budget $B$, let $\mathcal{Z}_B^{\text{Greedy}}$ be the set selected by our Algorithm 1, and $\mathcal{Z}_B^{\text{OPT}}$ be the optimal set that maximizes the regularized variant of problem (P3). The corresponding $f$ values of these sets are denoted by $f_B^{\text{Greedy}}$ and $f_B^{\text{OPT}}$ respectively.

**Theorem 1.** *Let $I_\ell^{\text{reg}}$ satisfies Assumption 1 and Assumption 2 requirements. Then, we have $f_B^{\text{Greedy}} \geq (1 - e^{-\gamma}) f_B^{\text{OPT}}$, where $\gamma = \frac{\kappa \cdot m_{2B + |\mathcal{G}|}}{M_{2B + |\mathcal{G}|}} \cdot \frac{(d_{\min}^{\text{non}})^2}{(d_{\max}^{\text{opt}})^2 + (d_{\min}^{\text{non}})^2}$.*

*Proof.* If $f$ is $\gamma$-weakly submodular at the set $\mathcal{Z}_B$ and the integer $B$ (i.e., $\gamma_{\mathcal{Z}_B, B} \geq \gamma$), then, using Theorem 3 from [29] (which holds for any normalized, monotone, $\gamma$-weakly submodular function), we can complete the proof of Theorem 1:

$$f^{\text{Greedy}} \geq \left(1 - e^{-\gamma_{\mathcal{Z}_B^{\text{Greedy}}, B}}\right) f^{\text{OPT}} \geq \left(1 - e^{-\gamma}\right) f^{\text{OPT}}.$$

Thus, it remains to prove the weak submodularity of $f$. Let $f_0$ denote $f$ with $\lambda = 0$, and define $\bar{D}(\mathcal{Z}) := D(\mathcal{Z} \cup \mathcal{G}) - D(\mathcal{G})$. Note that $\bar{D}$ is a normalized, monotone, submodular function. Then, the submodularity ratio of $f$ with general $\lambda$ is bounded as follows:

$$\gamma_{\mathcal{X},\mathcal{Y}} = \frac{\sum_{j \in \mathcal{Y}}(f_0(\mathcal{X} \cup \{j\}) - f_0(\mathcal{X})) + \lambda \sum_{j \in \mathcal{Y}}(\bar{D}(\mathcal{X} \cup \{j\}) - \bar{D}(\mathcal{X}))}{f_0(\mathcal{X} \cup \mathcal{Y}) - f_0(\mathcal{X}) + \lambda(\bar{D}(\mathcal{X} \cup \mathcal{Y}) - \bar{D}(\mathcal{X}))}$$

$$\geq \min\left(\frac{\sum_{j \in \mathcal{Y}}(f_0(\mathcal{X} \cup \{j\}) - f_0(\mathcal{X}))}{f_0(\mathcal{X} \cup \mathcal{Y}) - f_0(\mathcal{X})}, 1\right),$$

where the inequality is due to the fact that the submodularity ratio of $\bar{D}$ is $\geq 1$ [29]. If the submodularity ratio of $f_0$ is $\geq 1$, then $\gamma_{\mathcal{X},\mathcal{Y}} \geq 1$. This would lead to the following bound:

$$f^{\text{Greedy}} \geq \left(1 - \frac{1}{e}\right) f^{\text{OPT}}.$$

If the submodularity ratio of $f_0$ is $\leq 1$ (this would be the case in general; thus, we consider this case in the theorem), then the submodularity ratio $\gamma_{\mathcal{X},\mathcal{Y}}$ of $f$ with general $\lambda$ is lower bounded by the submodularity ratio of $f_0$. By applying Lemma 1 with $\left(\mathcal{Z}_B^{\text{Greedy}}, B\right)$, we have that (since $\left|\mathcal{Z}_B^{\text{Greedy}}\right| = B$):

$$\gamma_{\mathcal{Z}_B^{\text{Greedy}},B} \geq \frac{\kappa \cdot m_{2B+|\mathcal{G}|}}{M_{2B+|\mathcal{G}|}} \cdot \frac{(d_{\min}^{\text{non}})^2}{(d_{\max}^{\text{opt}})^2 + (d_{\min}^{\text{non}})^2} =: \gamma.$$

This completes the proof. $\square$

The following lemma provides a lower bound on the submodularity ratio $\gamma_{\mathcal{Z},k}$ of $f_0$ (for any $\mathcal{Z}$ s.t. $|\mathcal{Z}| \leq B$, and $k \leq B$):

**Lemma 1.** *Let the regularized informativeness criterion $I_\ell^{\text{reg}}$ satisfies the Assumption 1 and 2. Then, for any set $\mathcal{Z}$ s.t. $\mathcal{Z} \subseteq \mathcal{S} \backslash \mathcal{G}$, $|\mathcal{Z}| \leq B$, and $k \leq B$, the submodularity ratio $\gamma_{\mathcal{Z},k}$ of $f_0$ is lower bounded by*

$$\gamma_{\mathcal{Z},k} \geq \frac{\kappa \cdot m_{|\mathcal{Z}|+|\mathcal{G}|+k}}{M_{|\mathcal{Z}|+|\mathcal{G}|+k}} \cdot \frac{(d_{\min}^{\text{non}})^2}{(d_{\max}^{\text{opt}})^2 + (d_{\min}^{\text{non}})^2}.$$

*Proof.* Since $I_\ell^{\text{reg}}$ is $m_{2B+|\mathcal{G}|}$-restricted strongly concave and $M_{2B+|\mathcal{G}|}$-restricted smooth on $\Omega_{2B+|\mathcal{G}|}$, we have that $I_\ell^{\text{reg}}$ is $m_{|\mathcal{Z}|+|\mathcal{G}|+k}$-restricted strongly concave and $M_{|\mathcal{Z}|+|\mathcal{G}|+k}$-restricted smooth on $\Omega_{|\mathcal{Z}|+|\mathcal{G}|+k}$ for any $\mathcal{Z}$ s.t. $|\mathcal{Z}| \leq B$, and $k \leq B$. In addition $I_\ell^{\text{reg}}$ is $\tilde{M}_{|\mathcal{Z}|+|\mathcal{G}|+1}$-restricted smooth on $\tilde{\Omega}_{|\mathcal{Z}|+|\mathcal{G}|+1}$ since $\Omega_{|\mathcal{Z}|+|\mathcal{G}|+k} \supseteq \tilde{\Omega}_{|\mathcal{Z}|+|\mathcal{G}|+k} \supseteq \tilde{\Omega}_{|\mathcal{Z}|+|\mathcal{G}|+1}$ (and $M_{|\mathcal{Z}|+|\mathcal{G}|+k} \geq \tilde{M}_{|\mathcal{Z}|+|\mathcal{G}|+k} \geq \tilde{M}_{|\mathcal{Z}|+|\mathcal{G}|+1}$).

Consider the two sets $\mathcal{X}, \mathcal{Y}$ such that $(\mathcal{X} \cup \mathcal{G}) \cap \mathcal{Y} = \emptyset$, $\mathcal{X} \subseteq \mathcal{Z}$, and $|\mathcal{Y}| \leq k$. We proceed by upper bounding the denominator and lower bounding the numerator of Eq. (11). Let $\bar{k} = |\mathcal{X}| + |\mathcal{G}| + k$. First, we apply Definition 1 with $x = R^{(\mathcal{X})}$ and $y = R^{(\mathcal{X} \cup \mathcal{Y})}$ (note that $(x, y) \in \Omega_{\bar{k}}$):

$$\frac{m_{\bar{k}}}{2}\left\|R^{(\mathcal{X} \cup \mathcal{Y})} - R^{(\mathcal{X})}\right\|_2^2 \leq I_\ell^{\text{reg}}(R^{(\mathcal{X})}) - I_\ell^{\text{reg}}(R^{(\mathcal{X} \cup \mathcal{Y})}) + \left\langle \nabla I_\ell^{\text{reg}}(R^{(\mathcal{X})}), R^{(\mathcal{X} \cup \mathcal{Y})} - R^{(\mathcal{X})}\right\rangle.$$

Rearranging and noting that $I_\ell^{\text{reg}}$ is monotone for increasing supports:

$$0 \leq I_\ell^{\text{reg}}(R^{(\mathcal{X} \cup \mathcal{Y})}) - I_\ell^{\text{reg}}(R^{(\mathcal{X})}) \leq \left\langle \nabla I_\ell^{\text{reg}}(R^{(\mathcal{X})}), R^{(\mathcal{X} \cup \mathcal{Y})} - R^{(\mathcal{X})}\right\rangle - \frac{m_{\bar{k}}}{2}\left\|R^{(\mathcal{X} \cup \mathcal{Y})} - R^{(\mathcal{X})}\right\|_2^2$$

$$\leq \max_{v : v_{(\mathcal{X} \cup \mathcal{Y} \cup \mathcal{G})^c} = 0}\left\langle \nabla I_\ell^{\text{reg}}(R^{(\mathcal{X})}), v - R^{(\mathcal{X})}\right\rangle - \frac{m_{\bar{k}}}{2}\left\|v - R^{(\mathcal{X})}\right\|_2^2.$$

Setting $v = R^{(\mathcal{X})} + \frac{1}{m_{\bar{k}}}\nabla I_\ell^{\text{reg}}(R^{(\mathcal{X})})_{\mathcal{X} \cup \mathcal{Y} \cup \mathcal{G}}$ that achieves the maximum above, we have

$$0 \leq I_\ell^{\text{reg}}(R^{(\mathcal{X} \cup \mathcal{Y})}) - I_\ell^{\text{reg}}(R^{(\mathcal{X})}) \leq \frac{1}{2m_{\bar{k}}}\left\|\nabla I_\ell^{\text{reg}}(R^{(\mathcal{X})})_{\mathcal{X} \cup \mathcal{Y} \cup \mathcal{G}}\right\|_2^2$$

$$= \frac{1}{2m_{\overline{k}}} \left( \left\| \nabla I_\ell^{\mathrm{reg}}(R^{(\mathcal{X})})_{\mathcal{X} \cup \mathcal{G}} \right\|_2^2 + \left\| \nabla I_\ell^{\mathrm{reg}}(R^{(\mathcal{X})})_{\mathcal{Y}} \right\|_2^2 \right),$$

where the last equality is due to $(\mathcal{X} \cup \mathcal{G}) \cap \mathcal{Y} = \emptyset$.

Next, consider a single state $j \in \mathcal{Y}$. The function $I_\ell^{\mathrm{reg}}$ at $R^{(\mathcal{X} \cup \{j\})}$ is larger than the function at any other $R$ on the same support. In particular, $I_\ell^{\mathrm{reg}}\left(R^{(\mathcal{X} \cup \{j\})}\right) \geq I_\ell^{\mathrm{reg}}(y_j)$, where $y_j := R^{(\mathcal{X})} + \frac{\kappa}{\tilde{M}_{|\mathcal{X}|+|\mathcal{G}|+1}} \nabla I_\ell^{\mathrm{reg}}(R^{(\mathcal{X})})_j$. Noting that $\left(x = R^{(\mathcal{X})}, y = y_j\right) \in \tilde{\Omega}_{|\mathcal{X}|+|\mathcal{G}|+1}$ and applying Definition 1:

$$I_\ell^{\mathrm{reg}}(R^{(\mathcal{X} \cup \{j\})}) - I_\ell^{\mathrm{reg}}(R^{(\mathcal{X})})$$

$$\geq I_\ell^{\mathrm{reg}}\left(R^{(\mathcal{X})} + \frac{\kappa}{\tilde{M}_{|\mathcal{X}|+|\mathcal{G}|+1}} \nabla I_\ell^{\mathrm{reg}}(R^{(\mathcal{X})})_j\right) - I_\ell^{\mathrm{reg}}(R^{(\mathcal{X})})$$

$$\geq \left\langle \nabla I_\ell^{\mathrm{reg}}(R^{(\mathcal{X})}), \frac{\kappa}{\tilde{M}_{|\mathcal{X}|+|\mathcal{G}|+1}} \nabla I_\ell^{\mathrm{reg}}(R^{(\mathcal{X})})_j \right\rangle - \frac{\tilde{M}_{|\mathcal{X}|+|\mathcal{G}|+1}}{2} \left\| \frac{\kappa}{\tilde{M}_{|\mathcal{X}|+|\mathcal{G}|+1}} \nabla I_\ell^{\mathrm{reg}}(R^{(\mathcal{X})})_j \right\|_2^2$$

$$= \frac{\kappa}{\tilde{M}_{|\mathcal{X}|+|\mathcal{G}|+1}} \left\| \nabla I_\ell^{\mathrm{reg}}(R^{(\mathcal{X})})_j \right\|_2^2 - \frac{\kappa^2}{2\tilde{M}_{|\mathcal{X}|+|\mathcal{G}|+1}} \left\| \nabla I_\ell^{\mathrm{reg}}(R^{(\mathcal{X})})_j \right\|_2^2$$

$$\geq \frac{\kappa}{2\tilde{M}_{|\mathcal{X}|+|\mathcal{G}|+1}} \left\| \nabla I_\ell^{\mathrm{reg}}(R^{(\mathcal{X})})_j \right\|_2^2.$$

Summing over all $j \in \mathcal{Y}$:

$$\sum_{j \in \mathcal{Y}} \left[ I_\ell^{\mathrm{reg}}(R^{(\mathcal{X} \cup \{j\})}) - I_\ell^{\mathrm{reg}}(R^{(\mathcal{X})}) \right] \geq \frac{\kappa}{2\tilde{M}_{|\mathcal{X}|+|\mathcal{G}|+1}} \sum_{j \in \mathcal{Y}} \left\| \nabla I_\ell^{\mathrm{reg}}(R^{(\mathcal{X})})_j \right\|_2^2$$

$$= \frac{\kappa}{2\tilde{M}_{|\mathcal{X}|+|\mathcal{G}|+1}} \left\| \nabla I_\ell^{\mathrm{reg}}(R^{(\mathcal{X})})_{\mathcal{Y}} \right\|_2^2.$$

Then, we have:

$$\gamma_{\mathcal{X},\mathcal{Y}} \geq \frac{\kappa \cdot m_{|\mathcal{X}|+|\mathcal{G}|+k}}{\tilde{M}_{|\mathcal{X}|+|\mathcal{G}|+1}} \cdot \frac{\left\| \nabla I_\ell^{\mathrm{reg}}(R^{(\mathcal{X})})_{\mathcal{Y}} \right\|_2^2}{\left\| \nabla I_\ell^{\mathrm{reg}}(R^{(\mathcal{X})})_{\mathcal{X} \cup \mathcal{G}} \right\|_2^2 + \left\| \nabla I_\ell^{\mathrm{reg}}(R^{(\mathcal{X})})_{\mathcal{Y}} \right\|_2^2}$$

$$= \frac{\kappa \cdot m_{|\mathcal{X}|+|\mathcal{G}|+k}}{\tilde{M}_{|\mathcal{X}|+|\mathcal{G}|+1}} \cdot \frac{1}{\frac{\left\| \nabla I_\ell^{\mathrm{reg}}(R^{(\mathcal{X})})_{\mathcal{X} \cup \mathcal{G}} \right\|_2^2}{\left\| \nabla I_\ell^{\mathrm{reg}}(R^{(\mathcal{X})})_{\mathcal{Y}} \right\|_2^2} + 1}$$

$$\overset{(i)}{\geq} \frac{\kappa \cdot m_{|\mathcal{X}|+|\mathcal{G}|+k}}{\tilde{M}_{|\mathcal{X}|+|\mathcal{G}|+1}} \cdot \frac{1}{\frac{(d_{\max}^{\mathrm{opt}})^2}{(d_{\min}^{\mathrm{non}})^2} + 1}$$

$$\overset{(ii)}{\geq} \frac{\kappa \cdot m_{|\mathcal{Z}|+|\mathcal{G}|+k}}{M_{|\mathcal{Z}|+|\mathcal{G}|+k}} \cdot \frac{1}{\frac{(d_{\max}^{\mathrm{opt}})^2}{(d_{\min}^{\mathrm{non}})^2} + 1},$$

where $(i)$ is due to $\left\| \nabla I_\ell^{\mathrm{reg}}(R^{(\mathcal{X})})_{\mathcal{X} \cup \mathcal{G}} \right\|_2^2 \leq (d_{\max}^{\mathrm{opt}})^2$ and $\left\| \nabla I_\ell^{\mathrm{reg}}(R^{(\mathcal{X})})_{\mathcal{Y}} \right\|_2^2 \geq |\mathcal{Y}| (d_{\min}^{\mathrm{non}})^2 \geq (d_{\min}^{\mathrm{non}})^2$ (see Assumption 2); and $(ii)$ is due to $m_{|\mathcal{X}|+|\mathcal{G}|+k} \geq m_{|\mathcal{Z}|+|\mathcal{G}|+k}$ and $M_{|\mathcal{Z}|+|\mathcal{G}|+k} \geq \tilde{M}_{|\mathcal{X}|+|\mathcal{G}|+k} \geq \tilde{M}_{|\mathcal{X}|+|\mathcal{G}|+1}$ (note that $1 \leq |\mathcal{Y}| \leq k$ and $1 \leq |\mathcal{X}| \leq |\mathcal{Z}|$). $\square$

# D Additional Details and Proofs for using State Abstractions (Section 3.5)

We present an extension of our EXPRD framework that is scalable to large state spaces by leveraging the techniques from state abstraction literature [32–34]. We use an abstraction $\phi : \mathcal{S} \to \mathcal{X}_\phi$, which is a mapping from high-dimensional state-space $\mathcal{S}$ to a low-dimensional latent space $\mathcal{X}_\phi$. Let $\phi^{-1}(x) := \{s \in \mathcal{S} : \phi(s) = x\}, \forall x \in \mathcal{X}_\phi$. We propose the following pipeline (called EXPRD-ABS):

1. By using the original MDP $\overline{M} = (\mathcal{S}, \mathcal{A}, T, \gamma, P_0, \overline{R})$ and the abstraction $\phi$, we construct an abstract MDP $\overline{M}_\phi = (\mathcal{X}_\phi, \mathcal{A}, T_\phi, \gamma, P_0, \overline{R}_\phi)$ as follows, $\forall x, x' \in \mathcal{X}_\phi, a \in \mathcal{A}$: $T_\phi(x'|x, a) = \frac{1}{|\phi^{-1}(x)|} \sum_{s \in \phi^{-1}(x)} \sum_{s' \in \phi^{-1}(x')} T(s'|s, a)$, and $\overline{R}_\phi(x, a) = \frac{1}{|\phi^{-1}(x)|} \sum_{s \in \phi^{-1}(x)} \overline{R}(s, a)$. We compute the set of optimal policies $\overline{\Pi}_\phi^*$ for the MDP $\overline{M}_\phi$.

2. We run our EXPRD framework on $\overline{M}_\phi$ with $\Pi^\dagger = \overline{\Pi}_\phi^*$, and the resulting reward is denoted $\widehat{R}_\phi$. The corresponding MDP is denoted by $\widehat{M}_\phi = (\mathcal{X}_\phi, \mathcal{A}, T_\phi, \gamma, P_0, \widehat{R}_\phi)$.

3. We define the reward function $\widehat{R}$ on the state space $\mathcal{S}$ as follows: $\widehat{R}(s, a) = \widehat{R}_\phi(\phi(s), a)$. The corresponding MDP is denoted by $\widehat{M} = (\mathcal{S}, \mathcal{A}, T, \gamma, P_0, \widehat{R})$.

In summary, the EXPRD-ABS pipeline is given by: $\overline{M} \to \overline{M}_\phi \to \widehat{M}_\phi \to \widehat{M}$.

Define $\epsilon_\phi := \min_{x \in \mathcal{X}_\phi} \min_{a \in \mathcal{A} \setminus \overline{\Pi}_{\phi,x}^*} \overline{\delta}_{\phi,\infty}^*(x, a)$, where $\overline{\delta}_{\phi,\infty}^*$ is the $\infty$-step optimality gap in the abstract MDP $\overline{M}_\phi = (\mathcal{X}_\phi, \mathcal{A}, T_\phi, \gamma, P_0, \overline{R}_\phi)$. For our analysis, we require the abstraction $\phi : \mathcal{S} \to \mathcal{X}_\phi$ to satisfy the following conditions:

- $\phi$ is $(\epsilon_{\overline{R}}, \epsilon_T)$-approximate model irrelevant abstraction [34] for the MDP $\overline{M} = (\mathcal{S}, \mathcal{A}, T, \gamma, P_0, \overline{R})$, i.e., $\forall s_1, s_2 \in \mathcal{S}$ where $\phi(s_1) = \phi(s_2)$, we have, $\forall a \in \mathcal{A}$: $|\overline{R}(s_1, a) - \overline{R}(s_2, a)| \le \epsilon_{\overline{R}}$, and $\sum_{x' \in \mathcal{X}_\phi} \left| \sum_{s' \in \phi^{-1}(x')} (T(s'|s_1, a) - T(s'|s_2, a)) \right| \le \epsilon_T$.

- The change in the transition dynamics $T$ during the compression-decompression process using the abstraction $\phi$ is very small, i.e., $\max_{s,a} \sum_{s'} \left| T(s'|s, a) - \frac{T_\phi(\phi(s')|\phi(s),a)}{|\phi^{-1}(\phi(s'))|} \right| \le \frac{(1-\gamma)^2 \epsilon_\phi}{2\gamma R_{\max}}$.

The following theorem shows that any optimal policy induced by the reward $\widehat{R}$ resulting from the EXPRD-ABS pipeline acts nearly optimal w.r.t. $\overline{R}$:

**Theorem 2.** *Let $\phi : \mathcal{S} \to \mathcal{X}_\phi$ satisfy the conditions discussed above. The original reward function $\overline{R}$, and the reward function $\widehat{R}$ output by the EXPRD-ABS pipeline satisfy the following:* $\max_s \left| \overline{V}_\infty^*(s) - \overline{V}_\infty^\pi(s) \right| \le \frac{2\epsilon_{\overline{R}}}{(1-\gamma)^2} + \frac{\gamma \cdot \epsilon_T \cdot R_{\max}}{2(1-\gamma)^3}, \forall \pi \in \widehat{\Pi}^*$, *i.e., any optimal policy induced by $\widehat{R}$ acts nearly optimal w.r.t. $\overline{R}$.*

*Proof.* Given an abstract policy $\pi : \mathcal{X}_\phi \to \mathcal{A}$ acting on $\mathcal{X}_\phi$, we define the lifted policy $[\pi]_{\uparrow M} : \mathcal{S} \to \mathcal{A}$ as $[\pi]_{\uparrow M}(s) := \pi(\phi(s)), \forall s \in \mathcal{S}$. Similarly, given a set of policies $\Pi = \{\pi : \mathcal{X}_\phi \to \mathcal{A}\}$, we define $[\Pi]_{\uparrow M} := \{[\pi]_{\uparrow M} : \pi \in \Pi\}$. We define an auxiliary MDP $\widetilde{M} = (\mathcal{S}, \mathcal{A}, \widetilde{T}, \gamma, P_0, \widetilde{R})$, where $\widetilde{R}(s, a) = \widehat{R}_\phi(\phi(s), a)$, and $\widetilde{T}(s'|s, a) = \frac{T_\phi(\phi(s')|\phi(s),a)}{|\phi^{-1}(\phi(s'))|}$.

**Step $\overline{M} \to \overline{M}_\phi$.** Since $\phi$ is $(\epsilon_{\overline{R}}, \epsilon_T)$-approximate model irrelevant abstraction, we have the following (see [34]):

$$\left| \overline{Q}_\infty^*(s, a) - \overline{Q}_{\phi,\infty}^*(\phi(s), a) \right| \le \frac{\epsilon_{\overline{R}}}{1-\gamma} + \frac{\gamma \cdot \epsilon_T \cdot R_{\max}}{2(1-\gamma)^2}, \quad \forall s \in \mathcal{S}, a \in \mathcal{A},$$

where $\overline{Q}_{\phi,\infty}^*$ is the optimal action value function of the MDP $\overline{M}_\phi$. Then, for any $\pi \in \left[ \overline{\Pi}_\phi^* \right]_{\uparrow \overline{M}}$, we have the following (see [61]):

$$\max_s \left| \overline{V}_\infty^*(s) - \overline{V}_\infty^\pi(s) \right| \le \frac{2}{1-\gamma} \cdot \max_{s,a} \left| \overline{Q}_\infty^*(s, a) - \overline{Q}_{\phi,\infty}^*(\phi(s), a) \right| \le \frac{2\epsilon_{\overline{R}}}{(1-\gamma)^2} + \frac{\gamma \cdot \epsilon_T \cdot R_{\max}}{2(1-\gamma)^3},$$

i.e., any optimal policy of $\overline{M}_\phi$, when lifted to $\mathcal{S}$, acts as a near-optimal policy in $\overline{M}$.

**Step $\overline{M}_\phi \to \widehat{M}_\phi$.** In the step 2 of our EXPRD-ABS pipeline, we set $\Pi^\dagger = \overline{\Pi}_\phi^*$. Our EXPRD framework ensures that any optimal policy for $\widehat{M}_\phi$ is also optimal in $\overline{M}_\phi$, i.e., $\widehat{\Pi}_\phi^* \subseteq \overline{\Pi}_\phi^*$. In addition, since $\Pi^\dagger = \overline{\Pi}_\phi^*$ and $\Pi^\dagger \subseteq \widehat{\Pi}_\phi^*$, we have that $\widehat{\Pi}_\phi^* = \overline{\Pi}_\phi^*$.

**Step $\widehat{M}_\phi \to \widetilde{M}$.** By the definition of $\widetilde{M}$, $\phi$ is a model irrelevant abstraction for $\widetilde{M}$. Thus, we have the following (see [34]):

$$\widetilde{Q}_\infty^*(s,a) \;=\; \widehat{Q}_{\phi,\infty}^*(\phi(s),a), \quad \forall s \in \mathcal{S}, a \in \mathcal{A}. \tag{12}$$

From the above equation, note that $\widetilde{\Pi}^* = \left[\widehat{\Pi}_\phi^*\right]_{\uparrow \widetilde{M}}$. Finally, we have that, for any $\pi \in \widetilde{\Pi}^*$:

$$\max_s \left|\overline{V}_\infty^*(s) - \overline{V}_\infty^\pi(s)\right| \;\leq\; \frac{2\epsilon_{\overline{R}}}{(1-\gamma)^2} + \frac{\gamma \cdot \epsilon_T \cdot R_{\max}}{2(1-\gamma)^3},$$

i.e., any optimal policy of $\widetilde{M}$ acts as a near-optimal policy in the original MDP $\overline{M}$.

**Optimality in $\widetilde{M}$.** Our EXPRD framework guarantees the following:

$$\widehat{Q}_{\phi,\infty}^{\pi^\dagger}(x, \pi^\dagger(x)) \;\geq\; \widehat{Q}_{\phi,\infty}^{\pi^\dagger}(x, a) + \epsilon_\phi, \quad \forall x \in \mathcal{X}_\phi, a \in \mathcal{A}\backslash\overline{\Pi}_{\phi,x}^*, \pi^\dagger \in \Pi^\dagger,$$

which can be rewritten as follows:

$$\widehat{Q}_{\phi,\infty}^*(\phi(s), \pi^\dagger(\phi(s))) \;\geq\; \widehat{Q}_{\phi,\infty}^*(\phi(s), a) + \epsilon_\phi, \quad \forall s \in \mathcal{S}, a \in \mathcal{A}\backslash\overline{\Pi}_{\phi,\phi(s)}^*, \pi^\dagger \in \Pi^\dagger.$$

From the above inequality and using (12), we have the following:

$$\widetilde{Q}_\infty^*(s, [\pi^\dagger]_{\uparrow\widetilde{M}}(s)) \;\geq\; \widetilde{Q}_\infty^*(s,a) + \epsilon_\phi, \quad \forall s \in \mathcal{S}, a \in \mathcal{A}\backslash \left[\overline{\Pi}_\phi^*\right]_{\uparrow\widetilde{M},s}, [\pi^\dagger]_{\uparrow\widetilde{M}} \in \left[\Pi^\dagger\right]_{\uparrow\widetilde{M}},$$

which can be rewritten as follows:

$$\widetilde{Q}_\infty^*(s, \pi^*(s)) \;\geq\; \widetilde{Q}_\infty^*(s,a) + \epsilon_\phi, \quad \forall s \in \mathcal{S}, a \in \mathcal{A}\backslash\widetilde{\Pi}_s^*, \pi^* \in \widetilde{\Pi}^*.$$

From the above inequality, for any deterministic policy $\pi \notin \widetilde{\Pi}^*$, we have (at least on one state $s \in \mathcal{S}$):

$$\widetilde{V}_\infty^*(s) \;=\; \widetilde{Q}_\infty^*(s, \pi^*(s)) \;\geq\; \widetilde{Q}_\infty^*(s, \pi(s)) + \epsilon_\phi \;\geq\; \widetilde{Q}_\infty^\pi(s, \pi(s)) + \epsilon_\phi \;=\; \widetilde{V}_\infty^\pi(s) + \epsilon_\phi,$$

i.e., $\max_s \left|\widetilde{V}_\infty^*(s) - \widetilde{V}_\infty^\pi(s)\right| \geq \epsilon_\phi$.

**Comparison $\widehat{M}$ vs. $\widetilde{M}$.** Now, we show that any deterministic optimal policy in $\widehat{M}$ is also optimal in $\widetilde{M}$, i.e., $\widehat{\Pi}^* \subseteq \widetilde{\Pi}^*$. Let $\max_{s,a}\left\|T(\cdot|s,a) - \widetilde{T}(\cdot|s,a)\right\|_1 = \beta_T$. Then, for any $\widehat{\pi} \in \widehat{\Pi}^*$ and $s \in \mathcal{S}$, we have:

$$\left|\widetilde{V}_\infty^*(s) - \widetilde{V}_\infty^{\widehat{\pi}}(s)\right| \;\leq\; \left|\widetilde{V}_\infty^*(s) - \widehat{V}_\infty^{\widehat{\pi}}(s)\right| + \left|\widehat{V}_\infty^{\widehat{\pi}}(s) - \widetilde{V}_\infty^{\widehat{\pi}}(s)\right| \;\leq\; \frac{2\gamma\beta_T R_{\max}}{(1-\gamma)^2} \;<\; \epsilon_\phi,$$

where the second last inequality is due to Lemma 3 and Lemma 4 from [36]. Then, from the optimality in $\widetilde{M}$, it must me the case that $\widehat{\pi} \in \widetilde{\Pi}^*$.

Finally, for any $\pi \in \widehat{\Pi}^*$, we have:

$$\max_s \left|\overline{V}_\infty^*(s) - \overline{V}_\infty^\pi(s)\right| \;\leq\; \frac{2\epsilon_{\overline{R}}}{(1-\gamma)^2} + \frac{\gamma \cdot \epsilon_T \cdot R_{\max}}{2(1-\gamma)^3},$$

i.e., any optimal policy of $\widehat{M}$ acts as a near-optimal policy in the original MDP $\overline{M}$. $\qquad\square$

# E  Additional Details and Results for ROOMSNAVENV (Section 4.1)

In this appendix, we expand on Section 4.1 and provide a more detailed description of the setup as well as additional results. Full implementation of our techniques is available in a Github repo as mentioned in Footnote 1.

Recall that the MDP for ROOMSNAVENV has $|\mathcal{S}| = 49$ states corresponding to cells in the grid-world and four actions given by $\mathcal{A} := \{\text{"up"}, \text{"left"}, \text{"down"}, \text{"right"}\}$. To refer to a specific state, we will use an enumeration scheme where the bottom-left cell is $s = 0$; the cell numbers increase going from left to right and bottom to top. With this convention, the top-right cell with the goal is $s = 48$, and four "gates" (cells that need to be crossed to go across rooms when navigating to the goal) correspond to states $\{9, 15, 19, 37\}$. In this MDP, we have one goal state $s = 48$, i.e., the set $\mathcal{G}$ in the problem (P3) is $\{48\}$. Furthermore, the original reward function has $\overline{R}(48, \text{"right"}) = R_{\max}$ and is 0 elsewhere.

**Additional details for the techniques evaluated.** Below, we describe different reward design techniques along with hyperparameters that are evaluated in this section. More concretely, we have:

(i) $\widehat{R}_{\text{ORIG}}$ simply represents the default reward function $\overline{R}$.

(ii) $\widehat{R}_{\text{PBRS}}$ is obtained via the PBRS technique based on Eq. 1, see Section 2.

(iii) $\widehat{R}_{\text{CRAFT}(B)}$ is designed manually based on the ideas discussed in Section 2. For selecting the states that we will assign non-zero rewards, we first develop a set function $D$ as described below after this list. Then, for a fixed budget $B$, we pick a set of top $B + |\mathcal{G}|$ states that maximize the value of the set function $D$. Then, we assign rewards to these picked states as follows: (a) for the $B$ states excluding $|\mathcal{G}|$ goal states, we assign a reward of $+1$ for one of the optimal action and $-1$ for others; (b) for $|\mathcal{G}|$ goal states, we assign the same rewards as $\overline{R}$. For the evaluation, we use $B = 5$ and denote the function as $\widehat{R}_{\text{CRAFT}(B=5)}$.

(iv) $\widehat{R}_{\text{PBRS-CRAFT}(B=5)}$ is obtained via the reward shaping technique from [42]. First, we compute the optimal state value function $\widehat{V}^*_\infty$ w.r.t. $\widehat{R}_{\text{CRAFT}(B=5)}$ designed above, i.e., we need to solve the task with the reward function $\widehat{R}_{\text{CRAFT}(B=5)}$. Then, we obtain the reward function $\widehat{R}_{\text{PBRS-CRAFT}(B=5)}$ using the PBRS technique based on Eq. 1 with the value function $\widehat{V}^*_\infty$ instead of the optimal value function $\overline{V}^*_\infty$ w.r.t. $\overline{R}$.

(v) $\widehat{R}_{\text{EXPRD}(B,\lambda\to\infty)}$ is the reward function designed by our EXPRD framework for a budget $B$ and an extreme setting of $\lambda \to \infty$. For this setting, the problem (P3) reduces to (P1) corresponding to the reward design with subgoals pre-selected by the function $D$—we use the same function $D$ that we used for $\widehat{R}_{\text{CRAFT}}$ above. For the evaluation, we use $B = 5$ and denote the designed reward function as $\widehat{R}_{\text{EXPRD}(B=5,\lambda\to\infty)}$. As discussed in Section 3, the budget $B$ here refers to the additional number of states that are allowed to be in $\text{supp}(R)$ along with the goal states $\mathcal{G}$ (see (P3)). Apart from hyperparameters $B$ and $\lambda$, EXPRD requires a choice of $\Pi^\dagger$, $\mathcal{H}$, and $I(R)$ – we discuss that below after this list.

(vi) $\widehat{R}_{\text{EXPRD}(B,\lambda=0)}$ is the reward function designed by our EXPRD framework for a budget $B$ and an important setting of $\lambda = 0$ where the problem (P3) reduces to (P2) corresponding to fully automated reward design without using any prior knowledge about the importance of states. For budget $B$, we consider values from $\{3, 5, |S|\}$ and denote the designed reward functions as $\widehat{R}_{\text{EXPRD}(B=3,\lambda=0)}$, $\widehat{R}_{\text{EXPRD}(B=5,\lambda=0)}$, and $\widehat{R}_{\text{EXPRD}(B=|S|,\lambda=0)}$. As stated above, the budget $B$ here refers to the additional number of states that are allowed to be in $\text{supp}(R)$ along with the goal states $\mathcal{G}$; the choice of $\Pi^\dagger$, $\mathcal{H}$, and $I(R)$ is discussed below.

Here we describe the set function $D$ used for computing $\widehat{R}_{\text{CRAFT}(B=5)}$ and $\widehat{R}_{\text{EXPRD}(B=5,\lambda\to\infty)}$. For the set function $D$, we used a simple modular function given by $D(\mathcal{Z}) := \sum_{s\in\mathcal{Z}} w_s$ where $w_s$ is a weight/score assigned to a state $s$ capturing its importance in terms of reward design. We used the following weights: $w_s = 2$ for $s = 48$ (the goal state); $w_s = 1$ for $s = 9$, $s = 15$, $s = 19$, and $s = 37$ (the four "gates"); $w_s = 0.5$ for $s = 8$, $s = 11$, $s = 29$, and $s = 32$ (centers of the four rooms); and $w_s = 0.1$ otherwise. Even though this function is simple, it captures the prior knowledge one expects to intuitively apply in practice. In general, one could learn such $D$ automatically using the techniques from [21–24].

| Reward $\widehat{R}$ | Sparseness $|\text{supp}(\widehat{R})|$ | Invariance property | | Informativeness $I(\widehat{R})$ | Convergence: #Episodes to % value | | |
|---|---|---|---|---|---|---|---|
| | | Eq. 13 | Eq. 14 | | 25% | 75% | 95% |
| $\widehat{R}_{\text{ORIG}}$ | 1 | 0.0009 | 0.0009 | $-0.1557$ | $1,688$ | $6,752$ | $20,570$ |
| $\widehat{R}_{\text{PBRS}}$ | 49 | 0.0009 | 0.0009 | 0.0000 | 3 | 5 | 15 |
| $\widehat{R}_{\text{CRAFT}(B=5)}$ | 6 | $-4.8366$ | $-0.1645$ | $-0.1122$ | 1010 | $\infty$ | $\infty$ |
| $\widehat{R}_{\text{PBRS-CRAFT}(B=5)}$ | 49 | 0.0009 | 0.0009 | $-0.0797$ | 35 | 79 | 146 |
| $\widehat{R}_{\text{EXPRD}(B=5,\lambda\to\infty)}$ | 6 | 0.0000 | 0.0010 | $-0.1070$ | 49 | 773 | $14,252$ |
| $\widehat{R}_{\text{EXPRD}(B=3,\lambda=0)}$ | 4 | 0.0000 | 0.0009 | $-0.0842$ | 177 | 474 | $1,514$ |
| $\widehat{R}_{\text{EXPRD}(B=5,\lambda=0)}$ | 6 | 0.0000 | 0.0009 | $-0.0709$ | 37 | 280 | 822 |
| $\widehat{R}_{\text{EXPRD}(B=|S|,\lambda=0)}$ | 49 | 0.0000 | 1.5147 | 0.0000 | 9 | 48 | 90 |

Figure 6: Results for ROOMSNAVENV. The designed reward functions are evaluated w.r.t. criteria of sparseness, invariance, informativeness, and convergence. Here, the invariance property is captured through two different notions stated in Eq. 13 and Eq. 14 (a negative value represents a violation in the invariance property). Convergence is measured w.r.t the number of episodes needed to get a specific % of the total expected reward, and are based on the convergence results in Figure 3a.

Apart from $B$ and $\lambda$, EXPRD requires us to specify $\Pi^\dagger$, $\mathcal{H}$, and $I(R)$. For the results reported in Figures 3 and 6, we use the following parameter choices for EXPRD: $\mathcal{H} = \{1, 4, 8, 16, 32\}$, $I(R)$ is given by Eq. 15, and the set $\Pi^\dagger$ contains only one policy from $\overline{\Pi}^*$. Later in this section, we also consider variations of $\mathcal{H}$ and $I(R)$, and report additional results in Figures 9 and 10.

**Results w.r.t. different criteria.** Next, we evaluate the above-mentioned designed reward functions w.r.t. criteria of sparseness, invariance, informativeness, and convergence. Sparseness is measured by $|\text{supp}(\widehat{R})|$, and informativeness is measured by $I(\widehat{R})$ that is used in the optimization problem (P3). Convergence is measured w.r.t. the number of episodes needed to get a specific % of the total expected reward, and is based on the convergence results in Figure 3a by taking various horizontal slices of the convergence plot. To measure the invariance property, we consider two different notions stated below:

$$\min_{\widehat{\pi}^* \in \widehat{\Pi}^*} \min_{s \in \mathcal{S}} \left( \overline{Q}^*_\infty(s, \widehat{\pi}^*(s)) - \overline{Q}^*_\infty(s, \overline{\pi}^*(s)) \right) \text{ for any } \overline{\pi}^* \in \overline{\Pi}^* \tag{13}$$

$$\min_{\pi \in \Pi^\dagger} \min_{s \in \mathcal{S}} \min_{a \in \mathcal{A} \setminus \overline{\Pi}^*_s} \left( \widehat{Q}^\pi_\infty(s, \pi(s)) - \widehat{Q}^\pi_\infty(s, a) \right) \tag{14}$$

The notion in Eq. (13) looks at one of the optimal policy $\widehat{\pi}^*$ w.r.t. $\widehat{R}$, and compares the gap in Q action values w.r.t. $\overline{R}$ – this quantity should be zero to ensure that none of the optimal policies w.r.t. $\widehat{R}$ is suboptimal w.r.t. $\overline{R}$. The notion in (14) is closer to the invariance constraint that we incorporate in the optimization problem of EXPRD – this quantity should be non-negative to ensure that none of the optimal policies w.r.t. $\widehat{R}$ is suboptimal w.r.t. $\overline{R}$.

In Figure 6, we compare the designed reward functions w.r.t. these different criteria. In the "Sparseness" column, the quantity $|\text{supp}(\widehat{R})|$ is $B + 1$ for $\widehat{R}_{\text{CRAFT}(B=5)}$, $\widehat{R}_{\text{EXPRD}(B=5,\lambda\to\infty)}$, $\widehat{R}_{\text{EXPRD}(B=3,\lambda=0)}$, and $\widehat{R}_{\text{EXPRD}(B=5,\lambda=0)}$ as the goal states $\mathcal{G}$ are included in the design. In the "Invariance property" columns, we see that $\widehat{R}_{\text{CRAFT}(B=5)}$ fails to satisfy the invariance property highlighting the well-known "reward bugs" that can arise in this approach and mislead the agent into learning suboptimal policies (see Section 2 and [2, 3]); this issue is further emphasized in the "Convergence" columns for $\widehat{R}_{\text{CRAFT}(B=5)}$, highlighting that the agent is stuck with a suboptimal policy.

The last three columns related to "Convergence" highlight that the informativeness criteria we use in the optimization problem is a useful indicator about the agent's convergence when learning from designed reward functions. Furthermore, EXPRD can provide an effective trade-off in sparseness and informativeness while maintaining invariance property and speed up the agent's convergence. Even for small budgets of $B = 3$ or $B = 5$, the reward functions $\widehat{R}_{\text{EXPRD}(3,\lambda=0)}$ and $\widehat{R}_{\text{EXPRD}(5,\lambda=0)}$ lead to substantial speedups in the agent's convergence in contrast to the original reward function $\overline{R}$. Figures 7f and 7g further highlights that the states picked by EXPRD are important – the Algorithm 1 automatically picked the "gates" in the design process.

**Visualizations of the designed reward functions.** Figure 7 below shows a visualization of the eight different designed reward functions – this visualization is a variant of the visualization shown in Figure 3, where only three reward functions were shown.

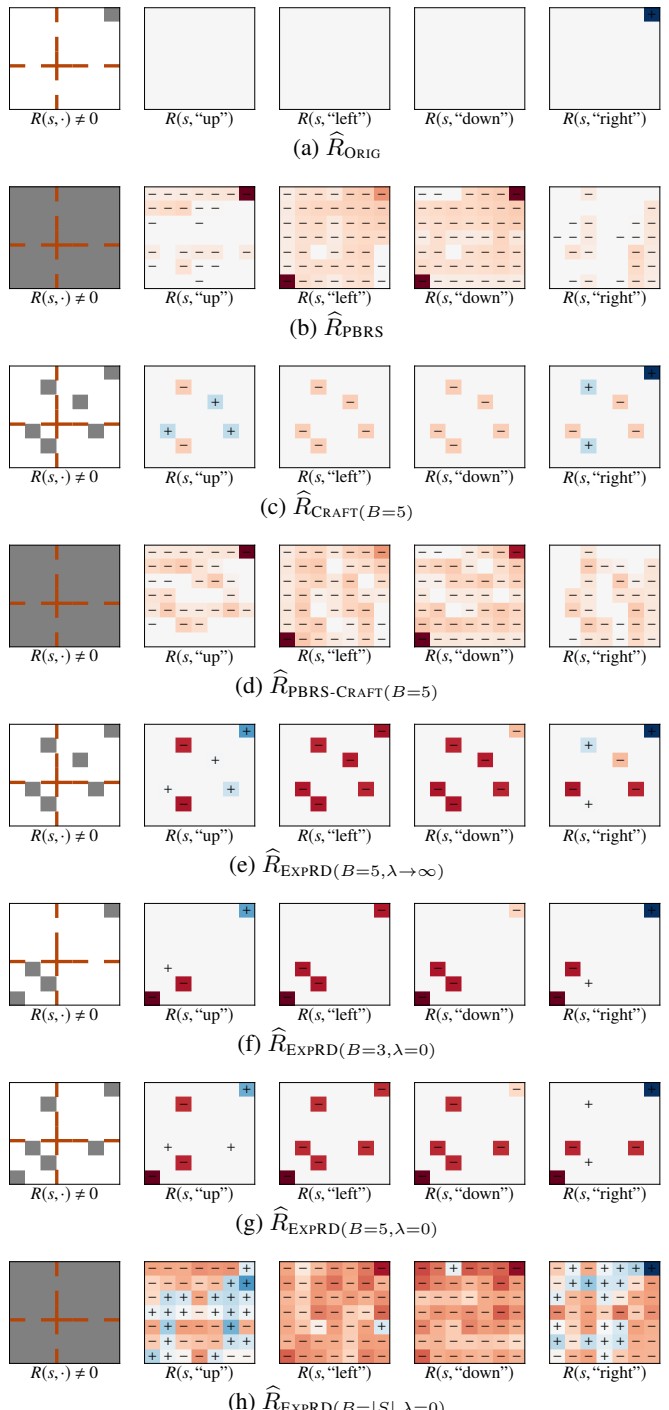

Figure 7: Results for ROOMSNAVENV. These plots show visualization of different designed reward functions discussed in Figure 6 – this visualization is a variant of the visualization shown in Figure 3 where only three reward functions were shown. For each of the reward functions, the first plot titled $R(s, .) \neq 0$ shows which states have a non-zero reward assigned to at least one action and are marked with Gray color. The next four plots titled $R(s, \text{"up"})$, $R(s, \text{"left"})$, $R(s, \text{"down"})$, $R(s, \text{"right"})$ show rewards assigned to each state/action: here, a negative reward is shown in Red color with sign "−", a positive reward is shown in Blue color with sign "+" and zero reward is shown in white. The magnitude of the reward is indicated by Red or Blue color intensity (see color representation in Figure 3).

**Results w.r.t. variations in $I(R)$.** For the results reported in Figures 3 and 9a, we fix $\mathcal{H} = \{1, 4, 8, 16, 32\}$, the set $\Pi^\dagger$ contains only one policy from $\overline{\overline{\Pi}}^*$, and we use the following functional form for $I(R)$ corresponding to the negated hinge loss:

$$I_1(R) := \frac{1}{|\Pi^\dagger| \cdot |\mathcal{H}| \cdot |\mathcal{S}|} \cdot \sum_{\pi^\dagger \in \Pi^\dagger} \sum_{h \in \mathcal{H}} \sum_{s \in \mathcal{S}} \max_{a \in \mathcal{A} \backslash \overline{\Pi}^*_s} \left( - \max(0, \overline{\delta}^*_\infty(s) - \delta^{\pi^\dagger}_h(s, a)) \right) \qquad (15)$$

Here, we perform additional experiments to understand the effect of variations in $I(R)$ on the reward functions designed by EXPRD. In Figures 9b, 9c, and 9d, we consider the following different functional forms of $I(R)$ corresponding to the negated hinge loss, respectively:

$$I_2(R) := \frac{1}{|\Pi^\dagger| \cdot |\mathcal{H}| \cdot |\mathcal{S}|} \cdot \sum_{\pi^\dagger \in \Pi^\dagger} \sum_{h \in \mathcal{H}} \sum_{s \in \mathcal{S}} \max_{a \in \mathcal{A} \backslash \overline{\Pi}^*_s} \left( - \max(0, \overline{\delta}^*_\infty(s, a) - \delta^{\pi^\dagger}_h(s, a)) \right) \qquad (16)$$

$$I_3(R) := \frac{1}{|\Pi^\dagger| \cdot |\mathcal{H}| \cdot |\mathcal{S}|} \cdot \sum_{\pi^\dagger \in \Pi^\dagger} \sum_{h \in \mathcal{H}} \sum_{s \in \mathcal{S}} \sum_{a \in \mathcal{A} \backslash \overline{\Pi}^*_s} \left( - \max(0, \overline{\delta}^*_\infty(s) - \delta^{\pi^\dagger}_h(s, a)) \right) \qquad (17)$$

$$I_4(R) := \frac{1}{|\Pi^\dagger| \cdot |\mathcal{H}| \cdot |\mathcal{S}|} \cdot \sum_{\pi^\dagger \in \Pi^\dagger} \sum_{h \in \mathcal{H}} \sum_{s \in \mathcal{S}} \sum_{a \in \mathcal{A} \backslash \overline{\Pi}^*_s} \left( - \max(0, \overline{\delta}^*_\infty(s, a) - \delta^{\pi^\dagger}_h(s, a)) \right) \qquad (18)$$

Finally, in Figures 9e and 9f, we use the following different functional forms of $I(R)$ corresponding to the linear and negated exponential functions (instead of negated hinge loss), respectively:

$$I_5(R) := \frac{1}{|\Pi^\dagger| \cdot |\mathcal{H}| \cdot |\mathcal{S}|} \cdot \sum_{\pi^\dagger \in \Pi^\dagger} \sum_{h \in \mathcal{H}} \sum_{s \in \mathcal{S}} \sum_{a \in \mathcal{A} \backslash \overline{\Pi}^*_s} \left( - (\overline{\delta}^*_\infty(s, a) - \delta^{\pi^\dagger}_h(s, a)) \right) \qquad (19)$$

$$I_6(R) := \frac{1}{|\Pi^\dagger| \cdot |\mathcal{H}| \cdot |\mathcal{S}|} \cdot \sum_{\pi^\dagger \in \Pi^\dagger} \sum_{h \in \mathcal{H}} \sum_{s \in \mathcal{S}} \sum_{a \in \mathcal{A} \backslash \overline{\Pi}^*_s} \left( - \exp(\overline{\delta}^*_\infty(s, a) - \delta^{\pi^\dagger}_h(s, a)) \right) \qquad (20)$$

Additionally, we report results by varying the choice of the set $\mathcal{H}$. More concretely, in Figure 10, we fix the functional form of $I(R)$ as given Eq. 15, the set $\Pi^\dagger$ is same as above, and we vary the set $\mathcal{H}$ as follows: $\{1, 4, 8, 16, 32\}$, $\{1, 2, \ldots, 19, 20\}$, and $\{10, 11, \ldots, 19, 20\}$. Note that the value 20 corresponds to $\frac{1}{1-\gamma}$.

All the results in this section are reported as an average over 40 runs and convergence plots show mean with standard error bars. Overall, the convergence behavior in Figures 9 and 10 suggests that the reward functions designed by our EXPRD framework are effective under different functional forms of $I(R)$ and different choices of the set $\mathcal{H}$.

**Run times for a varying number of states and actions.** Here, we report the run times for solving an instance of the optimization problem (P1) when set $\mathcal{Z}$ is fixed. In order to easily vary the number of states $|\mathcal{S}|$ as well as the number of actions $|\mathcal{A}|$, we consider a simple chain navigation environment where an agent can take "left" or "right" actions to navigate across the states (think of this as a one-dimensional variant of ROOMSNAVENV). To increase $|\mathcal{A}|$ beyond size 2, we added dummy actions which keep the agent's location unchanged. For reporting the run times, we consider $|\Pi^\dagger| = 1$, $\mathcal{H} = \{1, 4, 8, 16, 32\}$, and vary $|\mathcal{S}|$ as well as $|\mathcal{A}|$. These run times are reported when solving the formulation of the optimization problem in terms of matrices as shown in Section 2. Numbers are reported in seconds and are based on an average of 5 runs for each setting. These run times are obtained by running the computation on a laptop machine with 2.3 GHz Quad-Core Intel Core i5 processor and 16 GB RAM. Overall, these run times are of the same order as that of solving an optimization problem instance in environment poisoning attacks reported in the literature (see [19] and Section 5).

| $|\mathcal{A}|$ \\ $|\mathcal{S}|$ | 25 | 50 | 75 | 100 | 125 | 150 | 175 | 200 |
|---|---|---|---|---|---|---|---|---|
| 2 | 0.42s | 0.91s | 1.63s | 2.35s | 3.22s | 4.34s | 6.42s | 7.62s |
| 5 | 1.11s | 3.04s | 6.73s | 13.48s | 26.89s | 51.52s | 102.22s | 335.38s |

Figure 8: Run times for solving an instance of the optimization problem (P1) as we vary $|\mathcal{S}|$ and $|\mathcal{A}|$.

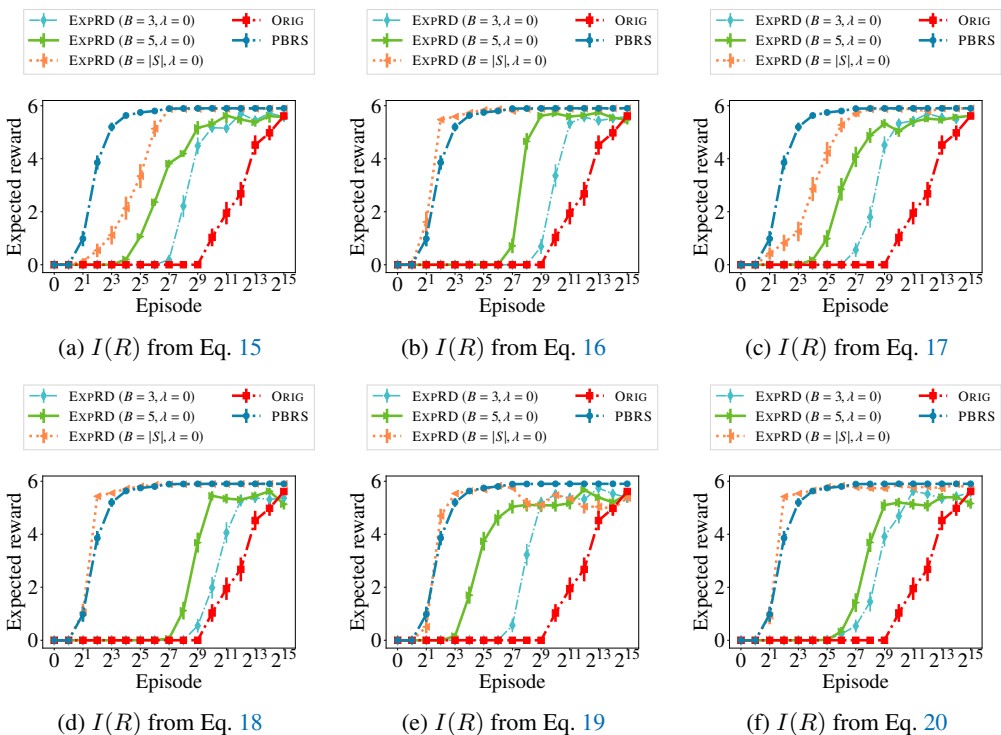

Figure 9: Results for ROOMSNAVENV. The plots show convergence in performance of the agent w.r.t. training episodes. Here, performance is measured as the expected reward per episode computed using $\overline{R}$; note that the x-axis is exponential in scale. As the parameter choices for EXPRD, we use $\mathcal{H} = \{1, 4, 8, 16, 32\}$ and the set $\Pi^{\dagger}$ contains only one policy from $\overline{\Pi}^{*}$. Each plot is obtained for a different functional form of $I(R)$.

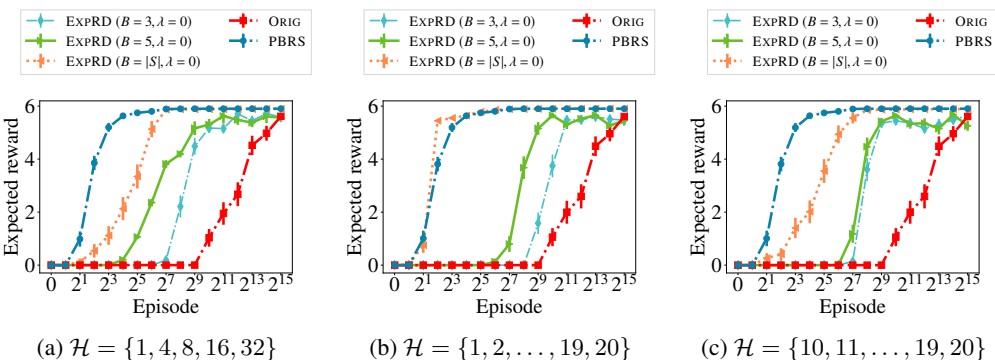

Figure 10: Results for ROOMSNAVENV. The plots show convergence in performance of the agent w.r.t. training episodes. Here, performance is measured as the expected reward per episode computed using $\overline{R}$; note that the x-axis is exponential in scale. As the parameter choices for EXPRD, we use $I(R)$ from Eq. 15 and the set $\Pi^{\dagger}$ contains only one policy from $\overline{\Pi}^{*}$. Each plot is obtained for a different choice of $\mathcal{H}$. Note that Figure 10a is same as Figure 9a.

## F    Additional Details and Results for LINEKEYNAVENV (Section 4.2)

In this appendix, we expand on Section 4.2 and provide a more detailed description of the setup as well as additional results. Full implementation of our techniques is available in a Github repo as mentioned in Footnote 1.

**Additional details for the techniques evaluated.**    Below, we describe different reward design techniques along with hyperparameters that are evaluated in this section. More concretely, we have:

(i) $\widehat{R}_{\text{ORIG}}$ simply represents the default reward function $\overline{R}$.

(ii) $\widehat{R}_{\text{PBRS}}$ is obtained via the PBRS technique based on Eq. 1 and using an abstraction (see Section 3.5, [35]). We first define an abstraction $\phi : \mathcal{S} \to \mathcal{X}_\phi$ as described below after this list. Based on this abstraction $\phi$, we construct an abstract MDP $\overline{M}_\phi$ using the original MDP $\overline{M}$, and compute the optimal state value function $\overline{V}^*_{\phi,\infty}$ in the abstract MDP $\overline{M}_\phi$. Finally, we lift $\overline{V}^*_{\phi,\infty}$ to the original state space $\mathcal{S}$ (see Appendix D), and use the lifted value function as the potential function for the PBRS.

(iii) $\widehat{R}_{\text{PBRS-ABS}}$ is a variant of $\widehat{R}_{\text{PBRS}}$. Similar to $\widehat{R}_{\text{PBRS}}$, we compute the optimal state value function $\overline{V}^*_{\phi,\infty}$ in the abstract MDP $\overline{M}_\phi$. We use this value function as the potential function for the PBRS to design $\widehat{R}_{\text{PBRS},\phi}$ in the MDP $\overline{M}_\phi$. Finally, we lift $\widehat{R}_{\text{PBRS},\phi}$ to the original state space $\mathcal{S}$ (see Appendix D). Note that $\widehat{R}_{\text{PBRS-ABS}}$ is not guaranteed to satisfy the invariance property of $\widehat{R}_{\text{PBRS}}$.

(iv) $\widehat{R}_{\text{EXPRD}(B,\lambda=0)}$ is the reward function designed by our pipeline in Section 3.5 that relies on our EXPRD framework and an abstraction. We use the same abstraction $\phi : \mathcal{S} \to \mathcal{X}_\phi$ for all the techniques and is described below after this list. In the subroutine, we run EXPRD on $\overline{M}_\phi$ for a budget $B = 5$ and a full budget $B = |\mathcal{X}_\phi|$; we set $\lambda = 0$. We denote the designed reward functions as $\widehat{R}_{\text{EXPRD}(B=5,\lambda=0)}$ and $\widehat{R}_{\text{EXPRD}(B=|\mathcal{X}_\phi|,\lambda=0)}$. Similar to Figure 9a, we fix $\mathcal{H} = \{1, 4, 8, 16, 32\}$, and we use the functional form given in Eq. 15 for $I(R)$.

Here, we describe the abstraction $\phi$ used for computing $\widehat{R}_{\text{PBRS}}$, $\widehat{R}_{\text{PBRS-ABS}}$, and $\widehat{R}_{\text{EXPRD}(B,\lambda=0)}$. Recall the description of the original MDP $\overline{M}$ from Section 4.2 – the state corresponds to the agent's status comprising of the current location (a point x in $[0, 1]$) and a binary flag whether the agent has acquired a key. For a given hyperparameter $\alpha \in (0, 1)$, we obtain a finite set of locations $X$ by $\alpha$-level discretization of the line segment $[0, 1]$, leading to a $1/\alpha$ number of locations. For the abstraction $\phi$ associated with this discretization, the abstract MDP $\overline{M}_\phi$ has $|\mathcal{X}_\phi| = 2/\alpha$ corresponding to $1/\alpha$ locations and a binary flag for the key. We use $\alpha = 0.05$ in the experiments.

**Results for Q-learning agent with** $0.01$**-level location discretization.**    For the results reported in the main paper (Figure 5a) and in Figure 11a, the agent uses Q-learning method in a discretized version of the original MDP $\overline{M}$ with a $0.01$-level discretization of the location (i.e., the number of states in the agent's discretized MDP is $200$). The rest of the method's parameters are same as in Section 4.1, i.e., we use standard Q-learning method for the agent with a learning rate $0.5$ and exploration factor $0.1$ [7]. During training, the agent receives rewards based on $\widehat{R}$, however, is evaluated based on $\overline{R}$. A training episode ends when the maximum steps (set to $50$) is reached or an agent's action terminates the episode. For this agent, the convergence results are reported in Figure 11a as an average over $40$ runs. These results demonstrate that all four designed reward functions—$\widehat{R}_{\text{PBRS}}$, $\widehat{R}_{\text{PBRS-ABS}}$, $\widehat{R}_{\text{EXPRD}(B=5,\lambda=0)}$, $\widehat{R}_{\text{EXPRD}(B=|\mathcal{X}_\phi|,\lambda=0)}$—substantially improves the convergence, whereas the agent is not able to learn under $\widehat{R}_{\text{ORIG}}$.

**Results for Q-learning agent with** $0.005$**-level location discretization.**    Here, we demonstrate that our abstraction based pipeline in Section 3.5 is robust to the state representation used by the agent. In particular, for the results reported in Figure 11b, the agent uses a discretized version of the original MDP $\overline{M}$ with a $0.005$-level discretization of the location. As in the setting above, the agent uses Q-learning method in this discretized version of the original MDP $\overline{M}$. Similar to Figure 11a, Figure 11b demonstrates that the performance associated with all four designed reward functions—$\widehat{R}_{\text{PBRS}}$, $\widehat{R}_{\text{PBRS-ABS}}$, $\widehat{R}_{\text{EXPRD}(B=5,\lambda=0)}$, $\widehat{R}_{\text{EXPRD}(B=|\mathcal{X}_\phi|,\lambda=0)}$—substantially improves the convergence in contrast to $\widehat{R}_{\text{ORIG}}$.

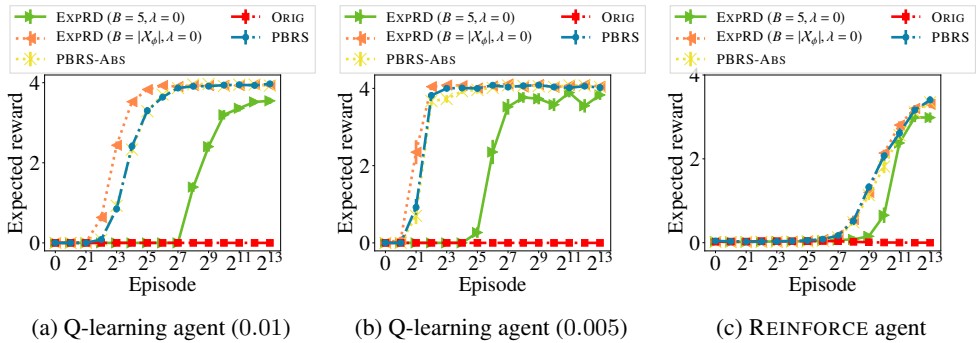

| (a) Q-learning agent (0.01) | (b) Q-learning agent (0.005) | (c) REINFORCE agent |

Figure 11: Results for LINEKEYNAVENV. These plots show convergence in performance of the agent w.r.t. training episodes. Here, performance is measured as the expected reward per episode computed using $\overline{R}$. **(a)** shows convergence for a Q-learning agent who uses a $0.01$-level discretization of the location. **(b)** shows convergence for a Q-learning agent who uses a $0.005$-level discretization of the location. **(c)** shows convergence for an agent who uses REINFORCE learning method with continuous representation of the location. All these agents receive rewards using the designed reward functions shown in Figure 12.

**Results for REINFORCE agent with continuous location representation.** For the results reported in Figure 11c, the agent uses the REINFORCE policy gradient method (see [7, 62]) in the original MDP $\overline{M}$ with continuous representation of the location. We use a neural network to learn the policy, which takes a continuous value in $[0, 1]$ (the location) and a binary flag (whether the agent has acquired a key) as the input representing a state $s$. The neural network has a hidden layer with $256$ nodes. Given a state $s$ (the input to the network), the policy network outputs three scores for three different actions. Then, applying softmax operation over these three scores gives the policy's action distribution. We use the REINFORCE method with a learning rate $0.0005$. The gradient update happens at the end of each episode. In contrast to the maximum episode length of $50$ used by Q-learning agents, we set this to $150$ for the REINFORCE agent.

Figure 11c shows convergence results for this agent as an average over $20$ runs; for each individual run, we additionally applied a moving-window average over a window size of $100$ episodes. With neural representation for states, the policy invariance might not hold anymore. However, Figure 11c demonstrates that all four designed reward functions—$\widehat{R}_{\text{PBRS}}$, $\widehat{R}_{\text{PBRS-ABS}}$, $\widehat{R}_{\text{EXPRD}(B=5,\lambda=0)}$, $\widehat{R}_{\text{EXPRD}(B=|\mathcal{X}_\phi|,\lambda=0)}$—substantially improves the convergence (slightly weaker compared to Figures 11a and 11b), whereas the agent is not able to learn under $\widehat{R}_{\text{ORIG}}$. This observation highlights our pipeline in Section 3.5 as a promising approach for reward design in high-dimensional settings. As future work, we plan to (both theoretically and empirically) investigate the effectiveness of the reward functions designed by our EXPRD framework or its adaptions in accelerating the learning process in high-dimensional settings for policy gradient methods.

**Visualizations of the designed reward functions.** Figure 12 shows visualization of the five different designed reward functions discussed above – this visualization is a variant of the visualization shown in Figure 5 where only three reward functions were shown. This visualization provides important insights into the reward functions designed by EXPRD. Interestingly, $\widehat{R}_{\text{EXPRD}(B=5,\lambda=0)}$ assigned a high positive reward for the "pick" action when the agent is in the locations with key (see $R((\text{x}, -), \text{"pick"})$ bar in Figure 12d).

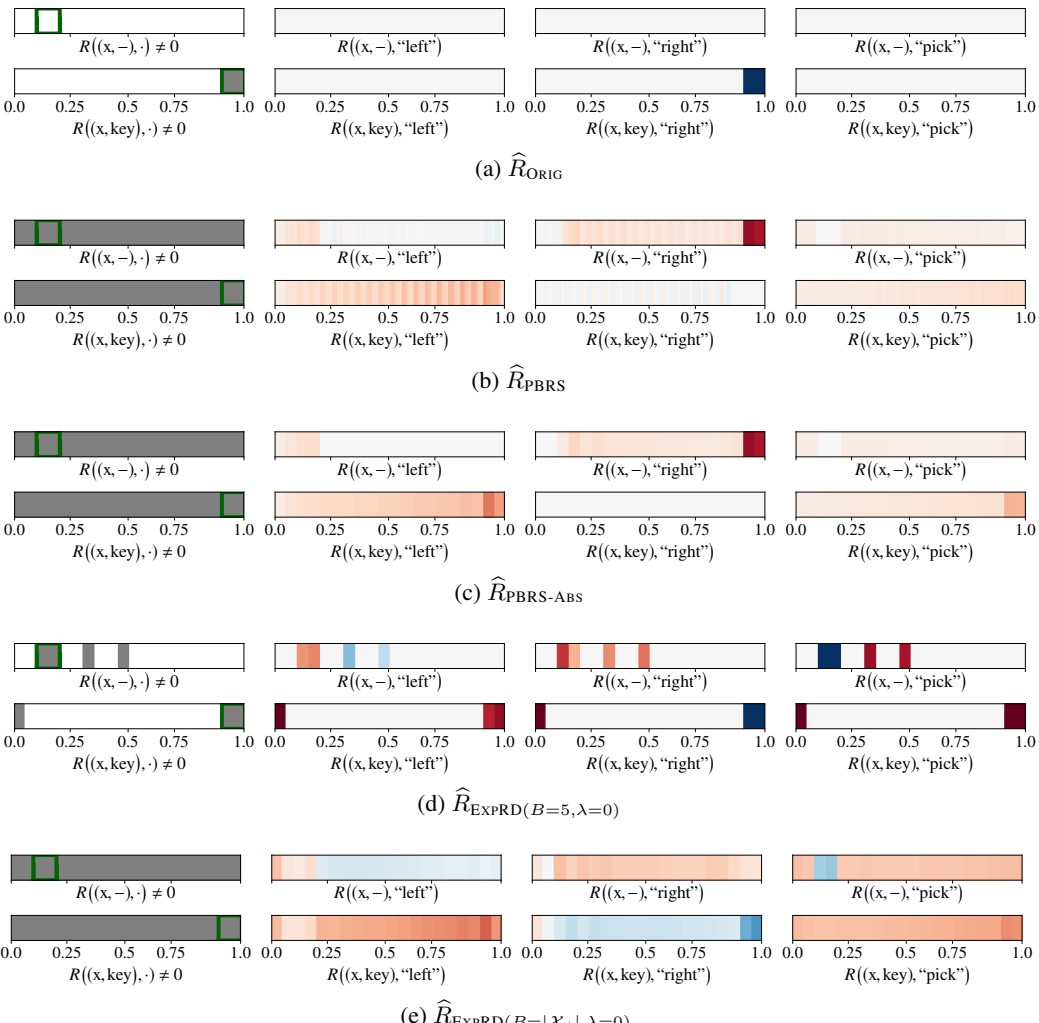

Figure 12: Results for LINEKEYNAVENV. These plots show visualization of the five different designed reward functions discussed above – this visualization is a variant of the visualization shown in Figure 5 where only three reward functions were shown. For each of the reward functions, we show a total of 8 horizontal bars. Denoting a state as tuple $(x, -)$ (i.e., location x when the key has not been picked) or $(x, key)$ (i.e., location x when the key has been picked), these 8 horizontal bars have the following interpretation. The two bars, titled $R((x, -), \cdot) \neq 0$ and $R((x, key), \cdot) \neq 0$, indicate states in Gray color for which a non-zero reward is assigned to at least one action; in these two bars, we have further highlighted the segment $[0.9, 1]$ with the goal, and the segment $[0.1, 0.2]$ with the key. The remaining six bars, titled $R((x, -), \text{"left"})$, $R((x, -), \text{"right"})$, $R((x, -), \text{"pick"})$, $R((x, key), \text{"left"})$, $R((x, key), \text{"right"})$, and $R((x, key), \text{"pick"})$, show rewards assigned to each state/action: here, a negative reward is shown in Red color, a positive reward is shown in Blue color, and zero reward is shown in white. The magnitude of the reward is indicated by Red or Blue color intensity and we use the same color representation as in Figure 5.