# OpenReview forum: "Explicable Reward Design for Reinforcement Learning Agents"
_NeurIPS.cc/2021/Conference — NeurIPS 2021 Poster_

### Official Review · Reviewer_jgko · 2021-07-15

**Rating:** 6
**Confidence:** 4

**Summary:**

This paper proposes an optimization-based framework for designing reward functions that balance sparsity/interpretability with informativeness. A novel optimization framework is proposed to directly learn these new reward signals given an initial reward signal and the corresponding optimal policy set. Three optimization problems are proposed that take into consideration subgoals, as well as learning subgoals on-the-fly given a budget on their number. An informativeness criterion is proposed based on Bellman residuals for optimization from which the new reward can be recovered. An iterative algorithm is proposed for approximately solving these problems, and state abstractions are also used to scale the method to larger state spaces. Theoretical analysis is provided to ensure feasibility of the optimization objective, and a near-optimal bound is provided for the iterative algorithm under reasonable assumptions on the informativeness criterion used. Finally experiments on simple domains are used to visualize the learned rewards in terms of their interpretability and informativeness.

**Limitations And Societal Impact:**

========  strengths  =========

1. Intelligent reward design is an important problem in the reinforcement learning community. The connection between sparsity and explainability is quite interesting and novel, and to my knowledge has not been studied in a mathematically coherent framework in the past. The paper is also very clearly written and the mathematics is precise.

2. The problem is cast as a problem in concave/convex optimization. This makes it possible to incorporate a variety of information into the objective function (e.g. subgoals or landmarks, other rewards, etc) and constrain the learned rewards appropriately. Theoretical results are provided that analyze the optimization objectives, proving feasibility. Furthermore, a greedy algorithm (Algorithm 1) is provided that iteratively solves these optimization problems near-optimally with a precise gap (Theorem 1).

3. The method, while inherently computationally challenging to solve in large MDPs, can be cast using state abstraction to reduce the dimensionality, and apply the proposed methods to this transformed MDP. The experiments, while toy in nature, demonstrate how the method works in discrete and a continuous environment. The visualization and interpretation of the learned behaviors is very well illustrated for these domains.

======== limitations ========

1. It is not immediately clear that sparseness of the reward function is the deciding criterion for interpretability. On the contrary, I believe it is not difficult to derive reward functions that are both dense and interpretable. For instance, "survivability" of the agent can be best modeled by assigning a (constant) positive reward on each transition, as seen in the classical cart-pole domain. In many robotics/nagivation problems, we could design a reward function that awards the agent based on how close the agent is to a pre-determined goal state. Finally, in many OR-related real-world problems, the reward is often best described in terms of revenues, e.g. for inventory control how many items are sold in a period. While I agree with you that sparsity can be useful for some toy domains, is it reasonable to expect similar notions on the type of problems above?

2. While the technical analysis of the method is very strong, a weaker point is the assumptions needed to realize the proposed methods. In lines 97-100, it is stated "learning an optimal policy induced by the sparse reward R-bar is challenging due to high sample complexity". The next lines mention that "... given R-bar and the corresponding optimal policy set". In some sense, it is difficult to see how these assumptions can be realized in practice on non-trivial domains, since learning this policy set is at least (if not more) challenging than learning even a single optimal policy with R-bar.

3. In light of point 2, I have to wonder if the initial motivation and assumptions, and experimental design could be tailored to solve a slightly different problem to the one initially proposed in the introduction, making the motivation of the paper much stronger. Namely, suppose we have solved a source task and obtained both an R-bar (learned from the source reward data) and a set of optimal policies. This knowledge can then be used to initialize the algorithm on a target environment in which the dynamics are different, but the goals remain the same. Since reward invariance - a key property the ExpRD method is designed to satisfy -- is required in this setup, does it mean it can be applied directly to landmark/subgoal discovery and transfer? If not, what could be changed in the methodology to allow it to work under change of dynamics?

======== other questions ========

1. Conditions (C1) and (C2) essentially require that the true Q-values are known. How can this condition be satisfied when they are estimated, up to some error say epsilon?

2. The current methodology seems quite reminiscent of the reward shaping literature in which arbitrary reward functions (in this case R-bar) can be converted to potential functions for PBRS (see, e.g. [1]). It seems as though subgoals can be encoded in a reward with "bugs", but then that method can be used to guarantee invariance, by learning a secondary value function on this subgoal-driven advice. This method has the advantage of being scalable to large or continuous spaces, and requiring little knowledge besides learning the secondary values on data. Given how similar the problem setting tackled by these two methods, can you compare them conceptually, if not experimentally? Does that method have limitations with regard to sparsity, or can sparsity be enforced in learning values by using state abstractions, sparse codings, etc?

3. Theorem 1 provides a gap between the optimal objective value and the greedy one. Is it possible to understand how "good" this gap is? Is it the best bound possible under the given assumptions?

[1] Harutyunyan, Anna, et al. "Expressing arbitrary reward functions as potential-based advice." Proceedings of the AAAI Conference on Artificial Intelligence. Vol. 29. No. 1. 2015.

**Main Review:**

originality: the work is original and novel, but some existing methodologies need further comparison, see limitations below

quality: the work is technically sound, and all claims are supported by proofs

clarity: the writing is very clear and precise, and well organized. however, I think it needs to be better motivated in the introduction, e.g. to state up-front what the use-cases of the method are, where it could be applied in practice, and where it could fail. This is somewhat indicated in the related work section, but I also feel could be elaborated on in the motivation. the assumptions of requiring the optimal policy set on the task seems to exclude a large number of use cases, which is important to discuss early on.

significance: the work is quite significant as reward design, sparsity and interpretability are all important topics currently studied in RL. it is not fully clear that the method is "better" than other methods, since not many have been discussed (see limitations below). the experiments only seem to consider simple baselines which does not answer this matter.

**Time Spent Reviewing:**

1

---

> ### Author Response · Authors · 2021-08-08
> **Response to Reviewer jgko**
>
> Thank you for carefully reviewing our paper! We greatly appreciate your feedback. Please see below our responses to your comments.
>
> -----
> **1. Notions of sparsity on the type of problems suggested in the review**
> - Thanks for the comments and the provided examples. We agree with the reviewer that beyond the specific sparsity structure we study, there are several other structural properties that serve as a deciding criterion for interpretability. For instance, it is also natural to consider reward functions with quantized values, parametric reward functions, and rewards with hierarchical or automata-based structures.
> - As future work, it would be very interesting to apply our ExpRD framework to the above-mentioned structural properties. In fact, one of the key significance of our framework is the information criterion $I(R)$ that is amenable to concave optimization for assigning reward values with a fixed reward structure; see Problem P1. When extending our framework to other structural properties, we would need to change Problems P2 and P3 appropriately. We will add this discussion in the revised paper.
> -----
> **2. Access to an optimal policy set vs. single optimal policy w.r.t. $\overline{R}$**
> - We would like to provide some thoughts on this issue in the context of constraints (C.2) in line 130. There are two quantities to consider here: (i) $\Pi^\dagger \subseteq \overline{\Pi}^*$ that determines the policy set used for enforcing invariance constraints, (ii) $\mathcal{A} \backslash \overline{\Pi}^*_s$ that denotes the set of suboptimal actions in state $s$ w.r.t. $\overline{R}$. We separately discuss these quantities in the next two points.
> - Regarding $\Pi^\dagger$: In all the experimental results reported in the paper (Section 4, Appendix E, and Appendix F), the set $\Pi^\dagger$ contains a single policy (see lines 278 and 780). We did try larger $\Pi^\dagger$ sets as well, but a singleton $\Pi^\dagger$ was enough to achieve similar results.
> - Regarding $\mathcal{A} \backslash \overline{\Pi}^*_s$: These sets can be obtained from the optimal action value function $\overline{Q}^*_\infty$.
> - In summary, our framework does not require an enumeration of optimal policy set; however, similar to PBRS with $\overline{V}^*_\infty$ as the potential function, our framework does require solving the task w.r.t. the original reward function $\overline{R}$. In the response below, we share further thoughts on this requirement.
>
> -----
> **3. Approximate $\overline{Q}^\*_\infty$ with $\epsilon$ error w.r.t. $\overline{R}$**
> - To apply the invariance constraints in (C.2), we note that the true action values $\overline{Q}^*_\infty$ are required to obtain the optimality gaps $\overline{\delta}_\infty^*(s,a)$, the sets of suboptimal actions $\mathcal{A} \backslash \overline{\Pi}^*_s$ in state $s$, and a subset of optimal policies $\Pi^\dagger$. When $\overline{Q}^*_\infty$ are only known up to an error $\epsilon$, we cannot directly apply the constraints in (C.2). Below, we provide some thoughts on how these constraints can be adapted for this setting.
> - Let us denote $\overline{Q}^{*,\epsilon}_\infty$ to be the $\epsilon$-approximate estimate of $\overline{Q}^\*_\infty$ w.r.t. the original reward function $\overline{R}$. First, we define a reference policy $\pi^\dagger$ using $\overline{Q}^{\*,\epsilon}_\infty$ as follows (with ties broken arbitrarily): $\pi^\dagger(s) := {\textrm{argmax}}_a \overline{Q}^{\*,\epsilon}_\infty (s,a).$ Then, we define the corresponding approximate optimality gaps as follows: $\overline{\delta}_\infty^{\*,\epsilon}(s,a) := \overline{Q}^{\*,\epsilon}_\infty (s,\pi^\dagger(s)) - \overline{Q}^{\*,\epsilon}_\infty (s,a).$ Finally, based on the above quantities, we adapt (C.2) as follows: $Q^{\pi^\dagger}_\infty(s, \pi^\dagger(s)) \geq  Q^{\pi^\dagger}_\infty(s, a) + \overline{\delta}_\infty^{\*,\epsilon}(s,a) - \epsilon’ \ \forall s \in \mathcal{S}, a \in \mathcal{A},$
> where $\epsilon’$ is a slack that can be adjusted based on the error $\epsilon$. We note that when $\epsilon = \epsilon’ = 0$, the modified (C.2) is a variant of the original (C.2) in the paper that still ensures the invariance property.
>
> -----
> **4. About Conditions (C.1) and (C.2) requiring the true Q-values**
> - We might have misinterpreted the reviewer’s comment, and we want to highlight two separate points here. First, we would like to clarify that the $Q^{\pi^\dagger}_\infty$ variables in (C.1) and (C.2) are just auxiliary variables in our optimization framework w.r.t. the designed reward function; see Problem P1. Second, we note that the true action values $\overline{Q}^\*_\infty$ are needed in (C.2), and our response above discusses the case when these values have errors.
>
> -----
> **5. Applying our framework in a transfer setting**
> - Thanks a lot for the suggestion! Reward shaping in a transfer setting is indeed very interesting. Below, we share a few thoughts on how one could use our framework for a transfer setting.
> - As an easier case, we can consider a transfer setting where the mismatch in dynamics is small and the optimal action values $Q$ in the source are $\epsilon$-approximate of the optimal action values $Q$ in the target. For this case, we can apply a relaxed formulation of the framework that we discussed in our response to comment 3. above.
> - As a more general case, we could possibly build on the transfer techniques of [Brys et al. 2015] and [Harutyunyan et al. 2015]. More concretely, we can use the source environment to obtain a new reward function $\widehat{R}^{src}$ using our framework. Then, $\widehat{R}^{src}$ can be useful in two ways. First, the $\textrm{supp}(\widehat{R}^{src})$ could capture a set of important states (subgoals) in the target environment which carry interpretable information. Second, the $\widehat{R}^{src}$ could be used to further design a PBRS-based reward function in the target to speed up the convergence -- we discuss this point further in the response below.
>
> [Harutyunyan et al. 2015] Harutyunyan et al. "Expressing arbitrary reward functions as potential-based advice." AAAI 2015.
>
> [Brys et al. 2015] Brys et al. "Policy transfer using reward shaping." AAMAS 2015.
>
> -----
> **6. Comparison with [Harutyunyan et al. 2015]**
> - Thanks a lot for the suggestion! Below, we share a few thoughts on the empirical and conceptual comparison of our work to the method in [Harutyunyan et al. 2015]. We will add this discussion in the revised paper.
> - Empirical comparison: Based on the reviewer's comment, we did an experiment by implementing a simplified variant of the reward shaping method proposed in [Harutyunyan et al. 2015]. In particular, we considered the RoomsNavEnv environment (Section 4.1; Appendix E) and used the "buggy" reward function $\widehat{R}_\textrm{CRAFT(B=5)}$ as an input for the method. Then, we computed its state value function and used it as a potential for shaping  -- let us denote the resulting reward function as $\widehat{R}_\textrm{PBRS-CRAFT(B=5)}$. In the context of Figure 6 (Appendix E), the time steps for $75$% convergence are $7$ for $\widehat{R}_\textrm{PBRS}$ and $89$ for $\widehat{R}_\textrm{PBRS-CRAFT(B=5)}$. In short, this method is very effective in terms of convergence. The sparseness-level of $\widehat{R}_\textrm{PBRS-CRAFT(B=5)}$ is similar to $\widehat{R}_\textrm{PBRS}$, as expected.
> -  Conceptual comparison: The main difference between our framework and the method in [Harutyunyan et al. 2015] is in terms of the sparseness structure of the designed reward functions. In this view, the reward function designed by the method in [Harutyunyan et al. 2015] is conceptually closer to the reward function $\widehat{R}_\textrm{PBRS}$. However, the method in [Harutyunyan et al. 2015] is more practical as it does not require solving the original task w.r.t. $\overline{R}$ that is very sparse.
> - Enforcing sparsity constraints in the method from [Harutyunyan et al. 2015]: One natural way to enforce sparsity in PBRS-based rewards is to make use of state abstractions [Marthi. 2007; Demir et al. 2019], and compute potential function on the abstracted states. However, one key challenge is in finding such meaningful abstractions (e.g., an automaton or an abstracted graphical structure of the MDP). Here, we believe techniques from subgoal discovery would be useful and provide an important direction for future work.
>
> [Marthi 2007]  Marthi. "Automatic shaping and decomposition of reward functions." ICML 2007.
>
> [Demir et al. 2019] Demir et al. "Landmark based reward shaping in reinforcement learning with hidden states." AAMAS 2019.
>
> -----
> **7. Quality of greedy solution in Theorem 1**
> - On the empirical side, we tried to compare the performance of our greedy solution with the optimal solution obtained via a brute force method, i.e., considering all possible subsets $\mathcal{Z}$ of budget $B$ and then solving Problem P1. However, we were only able to do the comparison for a small budget of $B=3$ (i.e., three additional states are allowed to be in $\textrm{supp}(R)$ along with the goal state). For this comparison, the performance of the reward function obtained from the greedy solution was close to that obtained via the brute force method. We will add this discussion and additional empirical results in the revised paper. On the theoretical side, it would be interesting to further quantify the quality of the gap.
>
> -----
> We hope that our responses can address your concerns and are helpful for improving your rating. We have also provided more experimental details, empirical results, and theoretical analysis in the appendix of the supplementary material. If you have any other comments or feedback, please let us know! We will be happy to provide further responses. We are looking forward to hearing back from you! Thank you again for the review.

---

> > ### Comment · Reviewer_jgko · 2021-08-22
> > **Still concerns regarding motivation**
> >
> > Thank you very much for your detailed responses. I believe that you have addressed most of my concerns about the technical contributions, which I believe are very strong in this paper, and the empirical results which you have summarized above.
> >
> > However, the main aspect that is still bothering me is whether there really is any meaningful connection between sparsity and interpretability, which I would still call into question in the general RL setting. I am wondering whether sparsity can be a useful property of a reward for other problems instead. For example, can sparsity be a better aspect to achieve when dealing with an adversary, who we wish will know very little about the problem we are trying to solve? I am also not sure entirely how sparse reward functions provide more information that can be used for debugging beyond grid-like domains, since (a) most real-world (non-goal) oriented problems don't quite have a sparse representation that is meaningful or interpretable, and (b) sparse reward problems could provide less information overall than dense ones, particularly for problems of sufficient complexity, making it harder to debug and understand what is learned. Overall, I am not questioning whether the trade-off has been optimized in the correct way, but I am not entirely convinced still that the trade-off is meaningful for a broad class of problems we are concerned with in practice.
> >
> > I could be willing to raise my score a little, considering that you have done a very good job of addressing my other concerns. I am still interested to see whether other reviewers now see the full merits of modeling this tradeoff, and there is something I may have missed here, and could adjust my score accordingly.

---

> > > ### Author Response · Authors · 2021-08-24
> > > **Response to Reviewer jgko (Part 2)**
> > >
> > > We sincerely thank the reviewer for providing valuable feedback. We are glad to hear that our responses have addressed many of your concerns. Regarding the reviewer’s concern about motivation, below we highlight the main motivating applications where sparsity structure in the reward design is important. More concretely, we provide further details about the three important practical settings that we introduced in the paper (lines 34-40).
> > >
> > >  - The first motivating application for our work is when rewards are designed for human agents learning to perform sequential tasks. For instance, we are interested in the application domain for educational simulators (e.g., a surgical training simulator) where a student needs to solve a complex task by accomplishing multiple milestones for success. Another application domain that inspires our work is tutoring systems for open-ended problem solving (e.g., block-based visual programming). In this context, a task can be quite complex with a goal-based reward; here, it is crucial to automatically identify intermediate milestones that could be useful for partial grading as feedback. Our goal is to design informative and interpretable reward functions for these educational applications. In future work, we plan to conduct user-studies to validate the importance of structured rewards in terms of interpretability for human participants. Our work provides an important stepping stone towards this goal.
> > >
> > > - The second motivating application for our work is when rewards are designed for complex compositional tasks that involve reward specifications in terms of logic, automata, or subgoals.  For instance, automata-driven reward specifications (or “reward machines”) are increasingly used in the robotics domain as a way to provide structured rewards. These structured rewards induce a form of sparsity on the underlying reward function. Given a fixed structure (e.g., an abstract logic formula without reward values), the “inner” Problem P1 in our framework can be used to instantiate this structure into an informative reward function. Furthermore, our framework provides a general recipe where the “outer” Problems P2 / P3 can be extended to choose a more suitable structure (e.g., a better logic formula). Thus, the sparsity structure considered in our work also provides an important basis for other structured rewards in complex tasks.
> > >
> > > - The third motivating problem setting for our work is related to adversarial attacks and defenses in RL. For instance, recent works have developed models of optimal reward-poisoning attacks​​; these attack models serve as a useful tool for finding security threats and designing novel algorithms robust to those threats. A popular attack model in literature is based on perturbing the reward function with Lp-norm (p>=1) constraints. As noted by the reviewer, our framework indeed provides a natural defense mechanism against such reward-poisoning attacks by enabling a system designer to construct a sparse reward function. Furthermore, we would like to note that Lp-norm (p>=1) attack models are not suitable for many real-world structured tasks (e.g., goal-based or compositional tasks). In this regard, our work provides a tool for developing new attack models that can optimize L0-norm constraints and is of interest to the adversarial RL research community. Here, an important research question is how to design defense mechanisms that make RL agents robust to L0-norm reward-poisoning attacks.
> > >
> > > Given these applications and problem settings, it is important to develop reward design frameworks that capture reward informativeness along with sparsity structure. Beyond the specific sparsity structure we studied, our work is an important step in the direction of reward design with structural constraints, which is an under-explored area. In particular, we believe our work makes two important technical contributions: (i) develops an optimization-based framework to design structured reward functions -- this allows us to go beyond using a potential function for designing rewards while satisfying some invariance property; (ii) introduces an information criterion to measure the quality of a given reward function that is amenable to concave optimization.
> > >
> > > We hope that the merits of our work could be appreciated by the reviewer, given the importance of the above-mentioned applications and the technical contributions. We sincerely hope that our responses can address your concerns and are helpful for improving your rating. If you have any other comments or feedback, please let us know! We will be happy to provide further responses. Thank you again for the review.

---

> > > > ### Comment · Reviewer_jgko · 2021-08-28
> > > > **I am more convinced now**
> > > >
> > > > Thank you for your detailed response and discussion about the motivations. When you frame the problem more generally as a trade-off between informativeness and sparsity, then I do agree that the paper has many interesting applications that can arise naturally. The technical insights in this paper are very strong, as the other reviewers have already pointed out. Upon reading the paper the first time, I felt that the key motivations and problem setting were not very well established (only one sentence listed possible applications, and it is hard to make the connection), which is much clearer now after reading your responses. I would recommend to revise the intro of the paper to make the motivation clearer, and showing a few illustrations could also be useful for readers. I am willing to raise my score.

---

> > > > > ### Author Response · Authors · 2021-08-28
> > > > > **Thank you for the feedback (Reviewer jgko)**
> > > > >
> > > > > We sincerely thank the reviewer for providing valuable and constructive feedback. As suggested by the reviewer, we will revise the introduction of the paper to make the motivation clearer and will expand on the discussion about the important practical settings. Also, we will add illustrations to provide a better perspective to the reader about the importance of the trade-off between informativeness and sparsity in the reward design.
> > > > >
> > > > > Furthermore, we will revise the paper as per our first response, in particular, adding the following details: (a) an empirical and conceptual comparison to the method in [Harutyunyan et al. 2015], (b) a discussion about possibly applying our framework in a transfer setting using techniques from [Brys et al. 2015], (c) a discussion about applying our framework when Q-values are only known with some error. We are grateful to the reviewer for the valuable feedback and it will definitely help in improving the paper. Thank you again for the review!

---

### Official Review · Reviewer_c46D · 2021-07-15

**Rating:** 7
**Confidence:** 3

**Summary:**

This paper tackles an important question in reward design for reinforcement learning - the trade-off between sparseness and learnability. It describes a framework to build modified rewards that are denser than the original but produce a set of optimal policies that are guaranteed to be a subset of the optimal policies induced by the original reward. The framework, ExpRD, is benchmarked on two relatively simple environments against several baselines. Experiments show improved convergence compared to the original sparse reward while being relatively sparser compared to PBRS.

**Limitations And Societal Impact:**

The authors point out that this paper is primarily a theoretical treatise on reward functions and that there are no direct negative social implications beyond risks inherent in any reward design. I would agree with the authors on this point.

**Main Review:**

This paper tackles an important question - how can one improve the informativeness of reward functions while not degrading their interpretability. To this question, the paper implies a strong correlation between sparseness and interpretability. Thus, the paper is fundamentally about the balance between informativeness and sparseness of reward functions. The authors take inspiration from PBRS - an important work that explored the design of dense rewards from an original sparse reward while avoiding locally optimal policies that are globally suboptimal (reward bugs). The ExpRD formulation achieves sparser rewards compared to PBRS.

Strengths:
1. The paper is grounded in theory - the authors provide detailed proofs of propositions in the appendix.
2. The paper provides a simple concave optimization method to derive sparse-but-informative rewards.
3. Provides a simple sparsity budget parameter B that can control the trade-off between sparseness and informativeness

Weaknesses:
1. The authors make "explicability" a central premise and imply an equivalence between explainability and sparsity. Beyond a few examples of rewards that are explainable and sparse, they do not explain why this equivalence should hold generally. Thus, this paper is fundamentally about the trade-off between how sparse and how learnable a reward function is.

2.  The paper does a mixed job of providing intuition behind the defined quantities or features. For example, the informativeness metric and its properties could have been better intuited.

3. Experiments conducted are relatively simple settings. While both environments capture the basic notion of a sparseness-informativeness trade-off, it is not clear that the performance improvements over the original sparse reward would hold for much larger state-spaces while still maintaining a comparable level of sparseness. The fundamental difference between PBRS and this method is the control over sparsity - and it is not clear to me that convergence would improve at all for small values of B when |S| is large.

Overall this is a good paper when viewed as a fundamental theoretical study of reward functions - but could be much stronger if the experimental treatment were to be more extensive.



**Time Spent Reviewing:**

2

---

> ### Author Response · Authors · 2021-08-08
> **Response to Reviewer c46D**
>
> Thank you for carefully reviewing our paper! We greatly appreciate your feedback. Please see below our responses to your comments.
>
> -----
> **1. This paper is fundamentally about the trade-off between how sparse and how learnable a reward function is.**
> - Thanks for the comments. We agree with the reviewer that the paper specifically focuses on sparsity structure, along with informativeness, in the reward design. We will make it more clear in the paper to avoid confusion for the readers.
> - Beyond sparsity, there are several other structural constraints that one could impose on reward functions (e.g., hierarchical, automata-based, or parametric structures). As future work, it would be very interesting to apply our ExpRD framework to these structural constraints on the reward functions. We want to highlight that one of the main contributions of the paper is the information criterion $I(R)$ itself as it is amenable to concave optimization in assigning rewards for a fixed reward structure; see Problem P1. When extending our framework to other structural constraints, we would need to appropriately change Problems P2 and P3, as well as algorithms (Section 3.3) and analysis (Section 3.4). We will add this discussion in the revised paper.
>
> -----
> **2. The paper does a mixed job of providing intuition behind the defined quantities or features. For example, the informativeness metric and its properties could have been better intuited.**
> - Thanks for the suggestion. We will expand the intuition behind the defined quantities and problem formulations in the revised paper. For example, we will add visual illustrations to explain the $h$-step optimality notions used in the definition of $I(R)$. We will also provide more intuition of the criterion $I(R)$ for a specific choice of $\ell$. Further, we will move Figure 6 (Appendix E) to the main paper as these results highlight different quantities for various designed reward functions.
>
> -----
> **3. Convergence for small values of $B$ when $|\mathcal{S}|$ is large**
> - Thanks for this question! Based on our empirical investigations with different settings, the convergence rate seems to be dictated by the bottleneck in exploration, i.e., how far the closest reward signal is for any state. For certain MDPs, as in LineKeyNavEnv (see Figure 11 in Appendix F), assigning reward values to a few informative states (in this case, the states with key) drastically speeds up the convergence. As the reviewer suggested, we agree that the budget $B$ would require some dependency on the MDP properties. We think that the dependency would not be on the size $|\mathcal{S}|$, but on some exploration-related properties of the MDP (e.g., the diameter of the MDP).
> - In general, it would be very interesting to formally analyze the trade-off between the budget $B$ and the convergence rate, and how this trade-off depends on the diameter of the MDP. Furthermore, it is useful to study connections between the budget $B$, the convergence rate, and the $h$-step optimality gaps in the information criterion. We will add this discussion in the revised paper.
>
> -----
> **4. Overall this is a good paper when viewed as a fundamental theoretical study of reward functions - but could be much stronger if the experimental treatment were to be more extensive.**
> -  Thanks for the comment. We have also provided more experimental details and empirical results in the appendix of the supplementary material (see Appendices E and F). We hope that these additional results can help alleviate the reviewer's concerns.
>
> -----
> We hope that our responses can address your concerns.  If you have any other comments or feedback, please let us know! We will be happy to provide further responses. We are looking forward to hearing back from you! Thank you again for the review.

---

> > ### Comment · Reviewer_c46D · 2021-08-27
> > **Connection to explainability is still tenuous**
> >
> > Thanks for the responses. I will retain my original rating as this is a reasonably good study of sparsity in reward functions. However, I will re-iterate that the paper appears to overstate the connection with explainability. Several other reviewers have also pointed out the same thing. I would encourage the authors to consider re-messaging to emphasize on learnability of a reward rather than explainability.

---

> > > ### Author Response · Authors · 2021-08-28
> > > **Thank you for the feedback (Reviewer c46D)**
> > >
> > > We sincerely thank the reviewer for the constructive feedback. We understand the concerns raised by reviewers, and as suggested, we will revise the paper appropriately. In particular, we will update the paper to emphasize how our framework trades off between the sparsity and learnability of a reward function. Also, as we mentioned in our response to other reviewers, we will revise the introduction of the paper to make the motivation clearer and will expand on the discussion about the important practical settings. We are grateful for all the feedback that will help in improving the paper. Thank you again for the review!

---

### Official Review · Reviewer_AMnw · 2021-07-16

**Rating:** 6
**Confidence:** 2

**Summary:**

The paper introduces ExpRD, a novel optimization framework to design explicable reward functions that attempts to balance informativeness and sparsity of the reward. The proposed reward design framework is theoretically underpinned into the existing framework of PBRS and tested in two distinct environments with some success.

**Limitations And Societal Impact:**

- The benefits of the proposed method hinges on sparsity being a preferred characteristic. I am not fully convinced of this. If the authors could clarify the case towards this, it would alleviate my primary reservation.

- Author indicates that the proposed method can be seen as a relaxation of PBRS towards (1) guaranteeing a weaker invariance property and (2) maximizing informativeness under (1) and given sparsity constraints. It is not clear to me what the redeemable benefit to this relaxation is as the empirical results do not seem to clearly indicate a marginal benefit.

- The work does not have any discernable negative societal impacts that is obvious to me.


**Main Review:**

- The paper is well-written and easy to follow.
- The related work section is extensive and provides a good overview of related literature on methods that address reward function design.

- I am not fully convinced that sparseness is a desirable characteristic for a reward function. I understand the reward functions are often sparse in real world environments and it is necessary to design learning algorithms that can work for them. However, the authors note sparsity as a “preferred characteristic” alongside informativeness [Line 100]. Settings highlighted to support this such as pedagogical and robotic applications [Line 35-38] might have to work with sparse rewards. However, I view this sparsity as a limitation to overcome rather than a characteristic to be desired. More discussion to support why we want sparsity would clarify the thesis of the paper.

- Results in 2a suggest that PBRS still outperforms all variants of the proposed method notwithstanding that the proposed method has more parameters that needs to be optimized. Results in 4a suggest something similar - the proposed method (one variant) outperforms PBRS only slightly. It isn’t clear whether this justifies the optimization necessary to acquire the parameters that achieves this for the proposed method while PBRS seems to work out of the box.


**Time Spent Reviewing:**

2

---

> ### Author Response · Authors · 2021-08-08
> **Response to Reviewer AMnw**
>
> Thank you for carefully reviewing our paper! Please see below our responses to your comments.
>
> -----
> **1. PBRS seems to work out of the box… The empirical results do not seem to clearly indicate a marginal benefit.**
> - Thanks for the comments. However, we feel that the reviewer has misinterpreted the general message conveyed through our experiments. We want to clarify these issues in the points below.
> - First, we would like to point out that the reward function $\widehat{R}_\textrm{PBRS}$ based on the PBRS technique using true value function $\overline{V}^*_\infty$ is *optimal*, i.e., it ensures that globally optimal actions for any state are also myopically optimal, thereby making the RL agent’s learning process trivial. In a setting with access to $\overline{V}^*_\infty$ (as in Figure 2a of Section 4.1), no other reward function can outperform $\widehat{R}_\textrm{PBRS}$ in terms of the RL agent’s convergence. In Figure 4a of Section 4.2, $\widehat{R}_\textrm{PBRS}$ performed slightly worse due to approximate estimation of $\overline{V}^*_\infty$ using state-abstraction.
> - Second, we would like to point out the following: While the PBRS technique (i.e., using a potential function for designing rewards) has laid out the theoretical foundations for reward design methods in the past two decades, these methods have to use a potential function for guaranteeing invariance. Our proposed theoretical framework allows one to go beyond a potential function, thereby opening new opportunities for designing reward functions through an optimization-based framework.
> - Third, coming back to the empirical evaluation, we want to highlight a few important points in the context of the RoomsNavEnv environment shown in Figures 2a (Section 4.1) and  Figures 5 / 6 (Appendix E). Comparison between $\widehat{R}_\textrm{CRAFT}(B=5)$ and $\widehat{R}_\textrm{ExpRD}(B=5,\lambda \to \infty)$ shows that the optimization problem P1 provides a principled way of assigning rewards values to a small set of preselected states while ensuring that the resulting reward function is not "buggy" and is informative for the convergence. For this common scenario of assigning rewards to preselected states (e.g., subgoals or landmarks), there is a very limited theoretical understanding of how to assign reward values without violating the invariance constraints. It is also important to note that the resulting reward function in $\widehat{R}_\textrm{ExpRD}(B=5,\lambda \to \infty)$ is not potential-based, and in fact, there might not exist any effective potential function that can be used to assign reward values to them for speeding up convergence.
> - Furthermore, comparison between $\widehat{R}_\textrm{ExpRD}(B=5,\lambda=0)$ and $\widehat{R}_\textrm{Orig}$ shows that the optimization problem P2 provides a principled way to automatic discovery of informative states along with assigning reward values to them. This is quite an interesting result on its own and shows that the informativeness criterion we have introduced is actually able to provide a suitable proxy for the RL agent’s convergence. We discuss this point further in the responses below.
>
> -----
> **2. Significance of the proposed reward design framework**
> - Building on the response above, here we want to provide further insights about the theoretical significance of the reward design framework introduced in the paper.  One of the important characteristics of our framework is allowing us to go beyond using a potential function for designing rewards while satisfying some invariance property. This brings us to the first important research question that we tackle in this paper: What is a suitable informativeness criterion to consider that could serve as a useful proxy for the RL agent’s convergence? One of our main contributions is the proposed information criterion $I(R)$ that measures the quality of a given reward function $R$ using the $h$-step optimality gaps. In recent years, there has been an increasing interest in designing such reward-centric information measures (see [Dai et al. 2019; Gleave et al. 2021; Furuta et al. 2021]). In particular, our information criterion $I(R)$ is amenable to concave optimization in assigning rewards for any fixed reward structure, as shown in Problem P1.
>
> - Second, while we focussed on sparsity as a structural constraint, we want to highlight that our work also provides a general recipe for developing an optimization-based reward design framework with different structural constraints. In particular, the “inner” Problem P1 can be seen as assigning reward values to a chosen reward structure that maximizes the information criterion $I(R)$; the “outer” Problem P2 / P3 can be seen as choosing a reward structure.  As future work, it would be very interesting to apply our ExpRD framework to different structural constraints on reward functions, as suitable for the applications. We will add this discussion in the revised paper.
>
> [Dai et al. 2019] Dai et al. “Maximum expected hitting cost of a Markov decision process and informativeness of rewards.” NeurIPS 2019.
>
> [Furuta et al. 2021] Furuta et al. “Policy information capacity: Information-theoretic measure for task complexity in deep reinforcement learning.” ICML 2021.
>
> [Gleave et al. 2021] Gleave et al. “Quantifying differences in reward functions.” ICLR 2021.
>
> -----
> **3. The importance of designing structured rewards and discussion of applications where sparsity is useful**
> - In this response, we would like to share a few thoughts on why structured reward design is an important research question, and also highlight specific applications where the sparsity structure is useful. We will add this discussion in the revised paper so that our contributions become more clear to the reader.
> - First, we believe that the desired characteristics of a reward function crucially depend on the application context. In the past few years, there has been an increasing interest in designing more structured reward functions, including automata-driven rewards [Camacho et al. 2017; Icarte et al. 2020], landmark-based sparse rewards [Demir et al. 2019], compositional rewards derived from logic [Jothimurugan et al. 2021], and hierarchical rewards [Krishnan et al. 2016]. The motivations for considering these different structures vary, including interpretability, transferability across tasks, and ease of distilling symbolic knowledge in the reward design process.
> - Second, the sparsity constraint is naturally tied to some of the above-mentioned structured rewards; furthermore, as discussed in the paper, there are several important applications where designing sparse rewards is desirable. More concretely, as discussed in lines 33-42, some form of sparsity structure is important in many practical settings, for instance, when rewards are designed for human learning agents and when rewards are designed for practitioners who then program these rewards into software as in robotics applications. Beyond the specific sparsity constraint under a budget, there are several other variants of sparsity properties that are also natural to consider, including reward functions with quantized values and automata-based rewards.
> - Crucially, given the above-mentioned applications and the growing interest in designing structured rewards, it is important to develop reward design frameworks that can capture reward informativeness under these structural constraints. Our proposed framework provides an important step in this direction of reward design with structural constraints, which is an under-explored area.
>
> [Krishnan et al. 2016] Krishnan et al. “Hirl: Hierarchical inverse reinforcement learning for long-horizon tasks with delayed rewards.” arXiv 2016.
>
> [Camacho et al. 2017] Camacho et al. “Decision-making with non-markovian rewards: From LTL to automata-based reward shaping.” RLDM 2017.
>
> [Demir et al. 2019] Demir et al. “Landmark based reward shaping in reinforcement learning with hidden states.” AAMAS 2019.
>
> [Icarte et al. 2020] Icarte et al. “Reward machines: Exploiting reward function structure in reinforcement learning.” arXiv 2020.
>
> [Jothimurugan et al. 2021] Jothimurugan et al. “Compositional reinforcement learning from logical specifications.” arXiv 2021.
>
> -----
> We hope that our responses can address your concerns and are helpful for improving your rating. We have also provided more experimental details, empirical results, and theoretical analysis in the appendix of the supplementary material. If you have any other comments or feedback, please let us know! We will be happy to provide further responses. We are looking forward to hearing back from you! Thank you again for the review.

---

> > ### Author Response · Authors · 2021-08-24
> > **Re: Response to Reviewer AMnw**
> >
> > We thank the reviewer again for their comments. We sincerely hope that our detailed responses have addressed your concerns and are helpful for improving your rating. If you have any other comments or feedback, please let us know! Thank you again for the review.

---

> > ### Comment · Reviewer_AMnw · 2021-08-30
> > **Re: Authors**
> >
> > Thank you for your detailed response clarifying the significance of the proposed method. Accordingly, I am raising my original rating. However, I do retain my reservation regarding the emphasis on sparsity and its connection with interpretability. As the other reviewers have also stated, I would encourage the authors to delineate this connection in the paper.

---

> > > ### Author Response · Authors · 2021-08-31
> > > **Thank you for the feedback (Reviewer AMnw)**
> > >
> > > We sincerely thank the reviewer for the valuable feedback. As suggested, we will revise the paper appropriately to tackle the concerns raised by reviewers. We are grateful for all the feedback that will help in improving the paper. Thank you again for the review!

---

### Official Review · Reviewer_5fTq · 2021-07-16

**Rating:** 7
**Confidence:** 2

**Summary:**

This paper the problem modifying a very sparse reward, e.g., an
indicator for task completion, in a manner that remains sparse but
increases the "informativeness". The resulting reward should ideally
result in faster convergence of RL style learning algorithms.  The
paper formulates this reward design problem as a form of discrete
optimization and provide a greedy approximation scheme, which is
proved to not deviate too much from the optimum.  Furthermore, a
theoretical analysis shows the relates this new measure to existing
reward design algorithms, and in particular potential based shaping.
Finally, the paper illustrates the approach on two domains, leveraging
state space abstraction to support continuous dynamics. There they show
that the they are indeed able to learn sparse rewards in a manner that
optionally incorporates and improves convergence time.



**Limitations And Societal Impact:**

I believe the problem of reward design is incredibly important for safe RL, both from an engineering perspective (less numerical errors and places to introduce the reward) and a human auditability perspective.

The primary limitation of this work, as I understand, seems to be that the reward designer must have a complete understanding of the dynamics. While reasonable for some applications, in many others, it somewhat defeats the purpose RL. Nevertheless, I think this is a very important subject to study.

**Main Review:**

I found the topic and techniques employed in this paper very
interesting. While I must admit it is outside my domain of
expertise, with the exception of the submodularity discussion,
I was able to follow the exposition. Again, since this is
not my primary area, I hesitate to comment on the novelty,
I believe the contribution to be original, with the paper
having a well made related work section.

This empirical evaluation highlights how useful the sparsity is
for interpretability, and I found it much easier to parse the
synthesized reward compared to the potential based approach.
Together with a strong theoretical basis, I find this to be
good motivation for accepting this work and allowing for a
more general discussion.

# Minor Points

A minor point of feedback and a point of early confusion on my part
was the mixing of stochastic and deterministic policies, sometimes
using the same notation, i.e., on line 97 is delta* for stochastic
policies?  And on line 92, is the set of policies deterministic?

Another point I found hard to initially follow was the motivation
for the particular form of the informative criterion. In particular,
how does the choice of the function l empirically and theoretically
affect convergence.

Finally, I think to me the most interesting feature this algorithm does not have is the ability to introduce new memory (state) which may make the reward even easier to interpret and potentially more robust to domain transfer - relying less on features of the dynamics.

**Time Spent Reviewing:**

5

---

> ### Author Response · Authors · 2021-08-08
> **Response to Reviewer 5fTq**
>
> Thank you for carefully reviewing our paper! We greatly appreciate your feedback. Please see below our responses to your comments.
>
> -----
> **1. Mixing of stochastic and deterministic policies**
> - Thanks for the comment. In lines 92 and 97, we are considering deterministic policies. This is mentioned in line 86: “From here onwards, we focus on deterministic  policies unless stated otherwise.” Also, as stated in line 84, focusing on deterministic policies is sufficient for our problem setting. We will clarify this further in the revised paper.
>
> -----
> **2. How does the choice of the function $I$ empirically and theoretically affect convergence**
> - Empirically, we did experiments with different choices of function $I$ and results are reported in Appendix E (see lines 811-827). In particular, we varied the choice of loss $\ell$, the functional form of $I$, and the choice of the set $\mathcal{H}$. We refer to the reviewer to Figures 8 and 9 in Appendix E.
> - It would also be very interesting to further analyze how the choice of $I$ affects the resulting reward function in terms of the RL agent’s convergence. This is an independent research question; in general, it is quite challenging to analyze reward-dependent convergence of RL agents, and there are limited results in this area. We will add a discussion about this point in the revised paper.
>
> -----
> **3. The most interesting feature this algorithm does not have is the ability to introduce new memory (state)**
> - Thanks for the suggestion! This is a very interesting idea to explore. One way to formulate this idea is to view it from the lens of creating landmarks / subgoals in the state space by adding new features in the state representation. We believe that this problem could also utilize theoretical tools from the discrete (submodular) optimization in choosing informative features.
> -  Another concrete way to formulate this problem is as follows: Consider a setting where we are given a (high-dimensional) feature representation for states and these features parametrize the reward function. For this setting, it would be interesting to develop a reward design framework that does sparse feature selection using tools from submodular optimization.
>
> -----
> We hope that our responses can address your concerns and are helpful for improving your rating or confidence score. We have also provided more experimental details, empirical results, and theoretical analysis in the appendix of the supplementary material. If you have any other comments or feedback, please let us know! We will be happy to provide further responses. We are looking forward to hearing back from you! Thank you again for the review.

---

### Author Response · Authors · 2021-08-08
**Response to all reviewers**

We thank all the reviewers for their careful reviews. Below, we provide responses to each reviewer separately.  We hope that our responses can help address reviewers’ concerns. If you have any other comments or feedback, please let us know. Thank you again for the reviews!

---

### Decision · Program_Chairs · 2021-09-27

**Decision:**

Accept (Poster)

**Comment:**

The reviewers agree that the problem of reward design is important and particularly relevant for safe reinforcement learning (RL). It is also a timely subject now that the deployment of RL systems in the real world is becoming increasingly more common. There was also a consensus among the reviewers that the proposed approach is technically sound.

Based on the submission, the reviews, the rebuttal and the subsequent discussions, we strongly advise the authors to make two modifications to the paper.

First, the placement of the proposed approach should be much more clear and stated at the outset. Some passages of the text suggest that the motivation underlying the proposed method is to find a reward to replace the original, sparse, reward in order to solve a specific task.

The connections made with Ng et al.'s PBRS and variants are one example. One of the requirements of the proposed approach is that one has access to (an approximation of) the optimal value function $\bar{V}^*$. One of the reviewers called attention to this fact and asked how realistic this is in practice, to which the authors responded that "similar to PBRS with $\bar{V}^*$ as the potential function, our framework does require solving the task w.r.t. the original reward function $\bar{R}$".

Although it is true that $\bar{V}^*$  can be used to derive PBRS' potential function, and Ng et al. point out that this choice gives rise to a particularly easy problem, their result apply to *any* potential function derived from a function $\Phi$ define over states (see Theorem 1 in Ng et al. 1999). In fact, Ng et al. emphasize that their method does *not* depend on $\bar{V}^*$, and also illustrate this point in their experiments by handcrafting a $\Phi$ that is intuitive but considerably different from $\bar{V}^*$.

PBRS and variants aim at finding a reward $\hat{R}$ to replace the original, potentially sparse, reward $\bar{R}$ in order to solve a specific task. If we view the proposed approach as an alternative to these methods, the fact that it needs $\bar{V}^*$ is a strong limitation.

An alternative view is that we *can* solve the task $\bar{R}$ but want a different version of it, $\hat{R}$, to be used in the future when we solve the task again. This new version of the reward, $\hat{R}$, should have two desirable properties: it should be informative and interpretable. Under this view, the fact that we need $\bar{V}^*$ seems to be less of an issue. Although this interpretation of the proposed approach seems to be favored in parts of the paper, it should be more clearly stated, and any passages that suggest otherwise should be modified to avoid ambiguities.

A second modification to the paper we strongly suggest regards the use of sparsity as a proxy for interpretability. This concern came up in most of the reviews and was a point of contention in the discussions. We advise the authors to add the good points made during the discussion to the paper. Specifically, they should clarify why sparsity is a good measure of interpretability and explain how the proposed approach can be modified to accommodate other reward structures. They should also add the provided list with examples of applications where sparsity structure in the reward design is important.

We hope the provided feedback will be helpful in making your paper even stronger.